# Decision Tree for Locally Private Estimation with Public Data

**Yuheng Ma**[1]    **Han Zhang**[1]    **Yuchao Cai**[2]    **Hanfang Yang**[13*]

[1]School of Statistics, Renmin University of China
[2]Faculty of Electrical Engineering, Mathematics and Computer Science, University of Twente
[3]Center for Applied Statistics, Renmin University of China
{yma,hanzhang0816,hyang}@ruc.edu.cn
y.cai@utwente.nl

## Abstract

We propose conducting locally differentially private (LDP) estimation with the aid of a small amount of public data to enhance the performance of private estimation. Specifically, we introduce an efficient algorithm called *Locally differentially Private Decision Tree* (LPDT) for LDP regression. We first use the public data to grow a decision tree partition and then fit an estimator according to the partition privately. From a theoretical perspective, we show that LPDT is $\varepsilon$-LDP and has a minimax optimal convergence rate under a mild assumption of similarity between public and private data, whereas the lower bound of the convergence rate of LPDT without public data is strictly slower, which implies that the public data helps to improve the convergence rates of LDP estimation. We conduct experiments on both synthetic and real-world data to demonstrate the superior performance of LPDT compared with other state-of-the-art LDP regression methods. Moreover, we show that LPDT remains effective despite considerable disparities between public and private data.

## 1 Introduction

Differential privacy (DP) [25] is a widely-used technique to protect sensitive information, like in medical trials [20], recommendation systems [49], and census data sharing [3]. Local differential privacy (LDP) [38, 24], a variation of DP, has gained particular attention, especially among industry experts [27, 35]. Unlike DP, LDP assumes data is privatized before being sent to a central collector. However, LDP models need more data to be accurate compared to DP [24], and many common techniques in data analysis such as standardization [10] and tree partition [59] are harder with LDP. This brings challenges to tasks such as density density estimation [23], mean estimation [24], and Gaussian estimation [36].

Fortunately, in some scenarios, private estimation performance can be enhanced with an additional public dataset [4, 7]. The public dataset can be either in-distribution, consisting of data from users who agree to share their personal information [6], or out-of-distribution, such as data from another source [34]. From a central DP perspective, an increasing amount of research has focused on leveraging public data to facilitate private learning, where public data mainly serves two purposes. On one hand, the knowledge learned from public data is implicitly transferred into the private model. Empirical investigations have demonstrated the effectiveness of pretraining on public data and fine-tuning privately on sensitive data [63, 43, 41, 61]. By gradient pre-conditioning with a subspace computed by public data, [65, 62, 37] managed to reduce the required amount of noise in differentially private gradient descent and accelerate its convergence. Through unlabeled public data, [45, 46] fed knowledge privately into student models. On the other hand, on public data, we can

---

*Corresponding author.

37th Conference on Neural Information Processing Systems (NeurIPS 2023).

conduct procedures that would be infeasible without access to the raw private data. For example, [10] used parameters computed by public data to standardize private data, which can augment the sample complexity of private mean estimation. Recently, [56] employed unlabeled public data to estimate the leading eigenvalue of the covariance matrix, resulting in an improved sample complexity for generalized linear models with non-interactive local differential privacy.

The paper focuses on the problem of non-parametric regression with LDP. While regression has been extensively studied in the central setting [50, 1, 14], the LDP case remains rarely explored. A notable reason is that most gradient-based methods [50, 1] are prohibited. In order to protect privacy, each data holder needs to compute the gradient of parameters locally, which requires a large amount of memory, computing power, and communication capacity on the terminal machine [52]. [29] proposed to impose Laplace noise on the data directly to provide privacy. However, this method is known to converge slowly [28] and suffer from the curse of dimensionality. More recently, [9, 33] investigated histogram-based approaches. Though theoretically optimal, histograms may perform poorly in practice, especially when the dimension of feature space is large. Thus, both methods proposed in [9, 33] face challenges when applied to real-world problems.

Under such background, using the idea of borrowing public data information, we propose an LDP non-parametric regression algorithm called the *Locally differentially Private Decision Tree* (LPDT) that achieves both optimal convergence rate and superior empirical performance. We first create a tree partition on the public dataset using the proposed *max-edge* rule. According to the partition, each data holder encodes the private data and releases the encoding which is processed using the proposed privacy mechanism. Finally, the curator aggregates the information in each partition cell and outputs a decision tree estimator. LPDT is advantageous from at least two perspectives: (i) LPDT integrates both benefits to leverage public data. It enables adaptive partitioning procedures which can eliminate some redundant cells and can transfer information from public data to private estimation through the tree partition. (ii) It inherits the merit of the decision tree model, such as interpretability, efficiency, stability, extensiveness to multiple feature types, and resistance to the curse of dimensionality.

We summarize our contributions. (i) For the first time, we propose to use public data in locally differentially private non-parametric regression. (ii) We propose a novel LDP regression algorithm called the locally differentially private decision tree that achieves theoretical optimality while maintaining satisfying practical performance. (iii) Under mild assumptions on the similarity between the distribution of public and private data, we establish the optimal convergence rate of LPDT, whereas the supremum of excess risk of LPDT without public fails to converge to zero. This demonstrates the theoretical advantage of incorporating public data. (iv) In experiments, we compare LPDT with other existing non-parametric LDP regression methods using both synthetic and real-world datasets. Our results demonstrate the overwhelming performance of LPDT, which illustrates the empirical improvement brought by public data. Moreover, we show that LPDT performs well even in the presence of significant disparities between public and private data.

## 2 Methodology

This section is dedicated to the methodology of LPDT. In Section 2.1, we first present notations and preliminaries related to regression problems, followed by a recap of the definition of local differential privacy. Next, we introduce our hybrid privacy mechanism for general partition-based estimation in Section 2.2. In Section 2.3, we propose our partition rule. Finally, in Section 2.4, we provide a comprehensive description of LPDT.

### 2.1 Preliminaries

**Notations** For any vector $x$, let $x^i$ denote the $i$-th element of $x$. Recall that for $1 \leq p < \infty$, the $L_p$-norm of $x = (x^1, \ldots, x^d)$ is defined by $\|x\|_p := (|x^1|^p + \cdots + |x^d|^p)^{1/p}$. Throughout this paper, we use the notation $a_n \lesssim b_n$ and $a_n \gtrsim b_n$ to denote that there exist positive constant $c$ and $c'$ such that $a_n \leq cb_n$ and $a_n \geq c'b_n$, for all $n \in \mathbb{N}$. In addition, we denote $a_n \asymp b_n$ if $a_n \lesssim b_n$ and $b_n \lesssim a_n$. Let $a \vee b = \max(a, b)$ and $a \wedge b = \min(a, b)$. Besides, for any set $A \subset \mathbb{R}^d$, the diameter of $A$ is defined by $\mathrm{diam}(A) := \sup_{x,x' \in A} \|x - x'\|_2$. Let the standard Laplace random variable have the continuous probability density function $p(x) = \frac{1}{2}e^{-|x|}$ for $x \in \mathbb{R}$.

Regression is to predict the value of an unobserved output variable $Y$ based on the observed input variable $X$, based on a dataset $D := \{(X_1, Y_1), \ldots, (X_n, Y_n)\}$ consisting of $n$ i.i.d. observations drawn from an unknown probability measure P on $\mathcal{X} \times \mathcal{Y} = [0,1]^d \times [-M, M]$. The density function of P is denoted as $s$. In addition, we have a public dataset $D^{pub} := \{(X_1^{pub}, Y_1^{pub}), \ldots, (X_{n_q}^{pub}, Y_{n_q}^{pub})\}$ drawn from distribution $Q$ on $\mathcal{X} \times \mathcal{Y}$ with sample size $n_q$. Its density function is denoted as $q$.

It is legitimate to consider the least square loss $L : \mathcal{X} \times \mathcal{Y} \times \mathcal{Y} \to [0, \infty)$ defined by $L(x, y, f(x)) := (y - f(x))^2$ for our target of regression. Then, for a measurable decision function $f : \mathcal{X} \to \mathcal{Y}$, the risk is defined by $\mathcal{R}_{L,\mathrm{P}}(f) := \int_{\mathcal{X} \times \mathcal{Y}} L(x, y, f(x)) \, d\mathrm{P}(x, y)$. The Bayes risk, which is the smallest possible risk with respect to P and $L$, is given by $\mathcal{R}_{L,\mathrm{P}}^* := \inf\{\mathcal{R}_{L,\mathrm{P}}(f)|f : \mathcal{X} \to \mathcal{Y} \text{ measurable}\}$. The function that achieves the Bayes risk is called Bayes function, namely, $f^*(x) := \mathbb{E}(Y|X = x)$.

**Definition 2.1 (Local Differential Privacy).** Given data $\{(X_i, Y_i)\}_{i=1}^n$, each $(X_i, Y_i)$ is mapped to a piece of privatized information $s_i$ which is a random variable on $\mathcal{S}$. Let $\sigma(\mathcal{S})$ be the $\sigma$-field on $\mathcal{S}$. $s_i$ is drawn conditional on $(X_i, Y_i)$ via the distribution R $(S \mid X_i = x, Y_i = y)$ for $S \in \sigma(\mathcal{S})$. Then the mechanism R provides $\varepsilon$-*local differential privacy* ($\varepsilon$-LDP) if

$$\sup\left\{\frac{\mathrm{R}\,(S \mid X_i = x, Y_i = y)}{\mathrm{R}\,(S \mid X_i = x', Y_i = y')} \mid S \in \sigma(\mathcal{S}), \text{ and } x, x' \in \mathcal{X}, \ y, y' \in \mathcal{Y}\right\} \le e^\varepsilon.$$

This formulation of local privacy is widely adopted [23, 9]. In contrast to central DP where the likelihood ratio is taken with respect to some statistics of all data, LDP requires individuals to guarantee their own privacy by considering the likelihood ratio with respect to each $(X_i, Y_i)$. Once the view $s$ is provided, no further processing can reduce the deniability about taking a value $(x, y)$ since any outcome $s$ is nearly as likely to have come from some other initial value $(x', y')$.

## 2.2 Privacy mechanism for tree partition

This section focuses on the hybrid privacy mechanism for general tree partitions. We first introduce the standard regression tree and then present our privacy mechanism based on the random response and Laplacian mechanism.

For index set $\mathcal{I}$, let $\pi = \{A_j\}_{j \in \mathcal{I}}$ be any tree partition of $\mathcal{X}$ with $\cup_{j \in \mathcal{I}} A_j = \mathcal{X}$ and $A_i \cap A_j = \emptyset$, $i \ne j$. For any $x \in \mathcal{X}$, let the cell containing $x$ be $A(x)$. A *population decision tree regressor* with partition $\pi$ is defined as

$$\overline{f}_\pi(x) = \sum_{j \in \mathcal{I}} \mathbf{1}\{x \in A_j\} \frac{\int_{A_j} f^*(x') \, d\mathrm{P}(x')}{\int_{A_j} d\mathrm{P}(x')}. \tag{1}$$

Here, we let $0/0 = 0$ by definition. To get a empirical estimator given the data set $D = \{(X_1, Y_1), \ldots, (X_n, Y_n)\}$, we estimate the numerator and the denominator of (1) respectively. To estimate the denominator, each sample $(X_i, Y_i)$ contributes a one-hot vector $U_i \in \{0,1\}^{|\mathcal{I}|}$ where the $j$-th element of $U_i$ is $\mathbf{1}\{X_i \in A_j\}$. Then an estimation of $\int_{A_j} d\mathrm{P}(x)$ is $\frac{1}{n} \sum_{i=1}^n U_i^j$, which is the number of samples in $A_j$ divided by $n$. Analogously, an estimation of $\int_{A_j} f^*(x) d\mathrm{P}(x)$ is $\frac{1}{n} \sum_{i=1}^n Y_i \cdot U_i^j$. Combining the pieces, a *decision tree regressor* is defined as

$$f_\pi(x) = \sum_{j \in \mathcal{I}} \mathbf{1}\{x \in A_j\} \frac{\sum_{i=1}^n Y_i \cdot U_i^j}{\sum_{i=1}^n U_i^j}. \tag{2}$$

In other words, $f_\pi(x)$ estimates $f(x)$ by the average of the responses in the cell $A(x)$. In the non-private setting, each data holder prepares $U_i$ and $Y_i$ according to the partition $\pi$ and sends it to the curator. Then the curator aggregates the transmission following (2).

To protect the privacy of each data, we propose to estimate the numerator and denominator of the population regression tree using a privatized method. Specifically, given $U_i^j$, we independently sample $\tilde{U}_i^j$ using the random response technique [57]

$$\tilde{U}_i^j = \begin{cases} U_i^j - \frac{1}{1+e^{\varepsilon/4}} & \text{with probability } \frac{e^{\varepsilon/4}}{1+e^{\varepsilon/4}} \\ 1 - U_i^j - \frac{1}{1+e^{\varepsilon/4}} & \text{with probability } \frac{1}{1+e^{\varepsilon/4}}. \end{cases} \tag{3}$$

Since $\mathbb{E}_{\mathrm{R}}\left[\frac{1}{n}\sum_{i=1}^n \tilde{U}_i^j\right] = \frac{e^{\varepsilon/4}-1}{e^{\varepsilon/4}+1}\frac{1}{n}\sum_{i=1}^n U_i^j$, we take $\frac{e^{\varepsilon/4}+1}{e^{\varepsilon/4}-1}\frac{1}{n}\sum_{i=1}^n \tilde{U}_i^j$ as the estimator of $\int_{A_j} d\mathrm{P}(x)$. To privatize $Y_1, \cdots, Y_n$, we use the standard Laplace mechanism [25]. Namely, we let

$$\tilde{Y}_i = Y_i + \frac{4M}{\varepsilon}\xi_i \tag{4}$$

where $\xi_i$ are i.i.d. standard Laplace random variables. Similarly, $\frac{e^{\varepsilon/4}+1}{e^{\varepsilon/4}-1}\frac{1}{n}\sum_{i=1}^n \tilde{Y}_i \cdot \tilde{U}_i^j$ can be used to estimate $\int_{A_j} f^*(x)d\mathrm{P}(x)$. Then using the privatized information $(\tilde{U}_i, \tilde{Y}_i), i = 1, \cdots, n$, we define the *locally differentially private decision tree regressor* as

$$f_\pi^{\mathrm{DP}}(x) = \sum_{j\in\mathcal{I}} \mathbf{1}\{x \in A_j\}\frac{\sum_{i=1}^n \tilde{Y}_i \cdot \tilde{U}_i^j}{\sum_{i=1}^n \tilde{U}_i^j}. \tag{5}$$

Compared to [9, 33] which used the Laplacian mechanism to protect both $U_i$ and $Y_i$, our mechanism (3) considers the fact that $U$ is a binary vector. When $|\mathcal{I}|$ is large, (3) can be more efficient than the Laplace mechanism which has a heavier-tailed distribution [23, 24].

## 2.3 Max-edge partition with variance reduction

While our privacy mechanism applies to any tree partition, it can be challenging to use general partitions such as the original CART [11] for theoretical analysis. Following the heuristic of [15], we propose a new splitting rule called the *max-edge partition rule* using the variance reduction criterion. This rule is amenable to theoretical analysis and can also achieve satisfactory practical performance. Given public dataset $\{(X_i^{pub}, Y_i^{pub})\}_{i=1}^{n_q}$, the partition rule is stated as follows:

- Let $A_0^1 := [0, 1]^d$ be the initial rectangular cell and $\pi_0 := \{A_0^j\}_{j\in\mathcal{I}_0}$ be the initialized cell partition. $\mathcal{I}_0 = \{1\}$ stands for the initialized index set. In addition, let $p \in \mathbb{N}$ represent the maximum depth of the tree and let $n_l$ represent the minimum sample size in each leaf. These parameters are fixed beforehand by the user and possibly depend on $n$.

- Suppose we have obtained a partition $\pi_{i-1}$ of $\mathcal{X}$ after $i-1$ steps of the recursion. Let $\pi_i = \emptyset$. In the $i$-th step, for each $A_{i-1}^j \in \pi_{i-1}, j \in \mathcal{I}_{i-1}$, suppose it is $\times_{\ell=1}^d[a_\ell, b_\ell]$. We choose the edge to be split among the longest edges. The index set of longest edges is defined as

$$\mathcal{M}_{i-1}^j = \left\{k \mid |b_k - a_k| = \max_{\ell=1,\cdots,d}|b_\ell - a_\ell|, \ k = 1, \cdots, d\right\}.$$

- Assume we split along the $\ell$-th dimension for $\ell \in \mathcal{M}_{i-1}^j$, $A_{i-1}^j$ is then partitioned into a left sub-cell $A_{i-1}^{j,0}(\ell)$ and a right sub-cell $A_{i-1}^{j,1}(\ell)$ along the midpoint of the chosen dimension, where $A_{i-1}^{j,0}(\ell) = \left\{x \mid x \in A_{i-1}^j, x^\ell < \frac{a_\ell+b_\ell}{2}\right\}$ and $A_{i-1}^{j,1}(\ell) = A_{i-1}^j/A_{i-1}^{j,0}(\ell)$. Then the dimension to be split is chosen using the variance reduction criterion:

$$\underset{\ell\in\mathcal{M}_{i-1}^j}{\arg\min}\ \sum_{i=1}^{n_q}\left(Y_i^{pub} - f_{\pi_{i-1}\cup A_{i-1}^{j,0}(\ell)\cup A_{i-1}^{j,1}(\ell)/A_{i-1}^j}(X_i^{pub})\right)^2. \tag{6}$$

- Once $\ell$ is selected, We count the number of samples in the sub-cells $\sum_{i=1}^n \mathbf{1}(X_i^{pub} \in A_{i-1}^{j,k}(\ell)), k = 0, 1$. If either of the cells contains fewer than $n_l$ samples, the splitting is pruned and we let $\pi_i = \pi_i \cup A_{i-1}^j$. Otherwise, let $\pi_i = \pi_i \cup \{A_{i-1}^{j,0}(\ell), A_{i-1}^{j,1}(\ell))\}$.

The complete process is presented in Algorithm 2 in the appendix. For each grid, the partition rule selects the midpoint of the longest edges that achieves the largest variance reduction. This procedure continues until there are not enough samples contained in any leaf node, or the depth of the tree reaches its limit.

## 2.4 Decision Tree with local differential privacy

With these preparations, we finally present the full procedure of LPDT in Algorithm 1.

**Algorithm 1:** Locally differentially private decision tree (LPDT)

---

**Input:** Private data $D = \{(X_i, Y_i)\}_{i=1}^n$, public data $D^{pub} = \{(X_i^{pub}, Y_i^{pub})\}_{i=1}^{n_q}$

**Parameters:** Depth $s$, minimum leaf sample size $n_l$.

   Curator create tree partition $\pi$ following max-edge rule in Section 2.3 on public data $D^{pub}$.

   Data holders of $D$ create privatized information (3) and (4) according to $\pi$.

   Curator aggregates the privatized information and compute $f_\pi^{\mathrm{DP}}$ by (5).

**Output:** The LPDT estimator $f_\pi^{\mathrm{DP}}$.

---

## 3 Theoretical results

In this section, we present our theoretical results and related comments. We first provide the $\varepsilon$-LDP guarantee of LPDT in Section 3.1. In Section 3.2, we establish the optimal convergence rate of LPDT with max-edge partition and the excess risk lower bound of LPDT without public data. Finally, we discuss the complexity of LPDT in Section 3.3.

### 3.1 Privacy guarantee for LPDT

**Theorem 3.1.** *Let $\pi = \{A_j\}_{j \in \mathcal{I}}$ be any partition of $\mathcal{X}$ with $\cup_{j \in \mathcal{I}} A_j = \mathcal{X}$ and $A_i \cap A_j = \emptyset$, $i \neq j$. Then the privacy mechanism $\mathrm{R}(\tilde{U}, \tilde{Y} | X, Y)$ defined in (3) and (4) is $\varepsilon$-LDP. Consequently, the LPDT estimator $f_\pi^{\mathrm{DP}}$ in Algorithm 1 is $\varepsilon$-LDP.*

### 3.2 Convergence rate of LPDT

We first present the necessary assumptions on the distribution P and Q.

**Assumption 3.2.** *Let $\alpha \in (0, 1]$. Assume the true regression function $f^* : \mathcal{X} \to \mathbb{R}$ is $\alpha$-Hölder continuous, i.e. there exists a constant $c_L > 0$ such that for all $x_1, x_2 \in \mathcal{X}$, $|f^*(x_1) - f^*(x_2)| \leq c_L \|x_1 - x_2\|^\alpha$. Also, assume that the density function of P is bounded, i.e. $p(x) \leq \bar{c}$ for some $\bar{c} > 0$.*

**Assumption 3.3.** *We assume that there exists some constant $\tau > 1$ such that for all cells $A \in \pi$, there holds $\tau^{-1} \int_A d\mathrm{Q}_X(x) \leq \int_A d\mathrm{P}_X(x) \leq \tau \int_A d\mathrm{Q}_X(x)$.*

Assumption 3.2 is a standard condition widely used in non-parametric statistics. Assumption 3.3 depicts the similarity between the distribution of public data and private data. It is also a mild assumption and requires only the probabilities in each cell under $\mathrm{P}_X$ and $\mathrm{Q}_X$ to be similar. When $p(x)$ and $q(x)$ are both bounded from 0, this assumption is satisfied. Alternatively, it suffices to require that $p(x)/q(x)$ is upper and lower bounded.

**Theorem 3.4.** *Let $f_\pi^{\mathrm{DP}}$ be the LPDT estimator in Algorithm 1. Suppose Assumption 3.2 and 3.3 hold. Then, for $n_q \gtrsim n^{\frac{d}{2\alpha+2d}}$, if we set $s \asymp \log n\varepsilon^2$ and $n_l \asymp n_q/2^s$, there holds*

$$\mathcal{R}_{L,\mathrm{P}}(f_\pi^{\mathrm{DP}}) - \mathcal{R}_{L,\mathrm{P}}^* \lesssim \left(\frac{\log n}{n\varepsilon^2}\right)^{\frac{\alpha}{\alpha+d} \wedge \frac{1}{3}}$$

*with probability $1 - 2/n_q^2 - 5/n^2$ with respect to $\mathrm{P}^n \otimes \mathrm{Q}^{n_q} \otimes \mathrm{R}^n$ where $\mathrm{R}^n$ is the joint distribution of privacy mechanisms in (3) and (4).*

Note that we only require $n_q \gtrsim n^{\frac{d}{2\alpha+2d}}$, which means the sample size of public data can be much smaller than private data. As illustrated in [33], the minimax convergence rate over Hölder function space is $(n(e^\varepsilon - 1)^2)^{-\frac{\alpha}{\alpha+d}}$, indicating that LPDT attains optimal rate when $\alpha/(\alpha + d) \leq 1/3$. In the case $\alpha/(\alpha + d) > 1/3$, or equivalently $2\alpha > d$, LPDT achieves fast yet sub-optimal convergence rate $n^{-\frac{1}{3}}$. Note that $2\alpha > d$ only when $d = 1$ and $\alpha \geq 1/2$, which rarely occurs. The next statement shows that LPDT fails without public data.

**Theorem 3.5.** *Let $f_\pi^{\mathrm{DP}}$ be the LPDT estimator in Algorithm 1 and $\mathcal{P}$ be the class of distributions satisfying Assumption 3.2. For $n_q = 0$ i.e. there is no public data, for any choice of $s$, $n_l$, and $\varepsilon$, there holds*

$$\sup_{\mathrm{P} \in \mathcal{P}} \left(\mathbb{E}\left[\mathcal{R}_{L,\mathrm{P}}(f_\pi^{\mathrm{DP}})\right] - \mathcal{R}_{L,\mathrm{P}}^*\right) \gtrsim 1.$$

Under the same hypothesis function space, the supremum of excess risk of LPDT does not even converge without public data. Together with Theorem 3.4, this shows that the prior information contained in public data can greatly enhance the quality of the private estimation.

We compare our results with those of others. LPDT converges faster than deconvolution-based method [29] whose rate is $n^{-\frac{2\alpha}{2\alpha+5d}}$ [28]. As for histogram-based methods, [9] achieves the optimal rate only when the density function is lower bounded, which is a strong condition. To avoid the condition, [33] derived an *ad hoc* estimator by adding a regularization to the marginal density estimation. LPDT takes another approach to avoid the condition. It does not apply any regularization or truncation on the estimator in each cell. Instead, as long as Assumption 3.3 holds, the low-density regions can be identified and treated with larger cells automatically by the parameter $n_l$. As a sacrifice, the large cells restrict the approximation ability of LPDT and the convergence rate is no more than $n^{-1/3}$. In addition, our theoretical results hold in the sense of "with high probability", which is more closely related to practical needs than "in expectation" as addressed in [9, 33].

Besides these advantages, we also discuss the benefit of public data for removing the range parameter [10, 9]. Consider the example from [33], where the convergence rate is given by $\left(\frac{r_n^{2d}}{n\varepsilon^2}\right)^{\frac{\alpha}{\alpha+d}}$. When the set $\mathcal{X} = \times_{j=1}^d [a^j, b^j]$ is unknown, it becomes necessary to create a histogram partition over $\times_{j=1}^d [-r_n, r_n]$, introducing an additional factor of $r_n^{2d}$. However, with publicly available data, we can approximate the range of the $j$-th dimension using $\widehat{a^j} = \min_i X_i^j$ and $\widehat{b^j} = \max_i X_i^j$. Subsequently, we can perform min-max scaling on each data point from $\times_{j=1}^d [\widehat{a^j}, \widehat{b^j}]$ to map it into $\times_{j=1}^d [0, 1]$, and then train an LPDT on $\times_{j=1}^d [0, 1]$. Any $x$ that falls outside the range $\times_{j=1}^d [0, 1]$ is predicted as 0. We demonstrate that by this approach, Theorem 3.4 holds with a probability of at least $1 - d/n_q^2$. See derivations in Section C.6 in the appendix.

### 3.3 Complexity analysis

We demonstrate that LPDT is an efficient method. We first consider the average computation complexity of LPDT. The training stage consists of two parts. The partition procedure takes $\mathcal{O}(sn_q d)$ time and the computation of (5) takes $\mathcal{O}(sn)$ time. From the proof of Theorem 3.4, we know that $2^s \asymp \left(n\varepsilon^2/\log n\right)^{-\frac{d}{2\alpha+2d}}$. Thus the training stage complexity is around $\mathcal{O}(n\log n\varepsilon^2 + n_q d\log n\varepsilon^2)$. Since each prediction of the decision tree takes $\mathcal{O}(s)$ time, the test time for each test instance is around $\mathcal{O}(\log n\varepsilon^2)$. As for storage complexity, since LPDT only requires the storage of the tree structure and the prediction value at each node, the space complexity of LPDT is $\mathcal{O}\left(\left(n\varepsilon^2/\log n\right)^{-\frac{d}{2\alpha+2d}}\right)$. In short, LPDT is an efficient method with a small number of parameters.

Table 1: Comparison of complexities of LDP regression methods.

|  | LPDT | PHIST [9] | DECONV [29] |
|---|---|---|---|
| Training Time Complexity | $\mathcal{O}(n\log n\varepsilon^2 + n_q d\log n\varepsilon^2)$ | $\mathcal{O}(nd\log n\varepsilon^2)$ | - |
| Testing Time Complexity | $\mathcal{O}(\log n\varepsilon^2)$ | $\mathcal{O}(\log n\varepsilon^2)$ | $\mathcal{O}(nd)$ |
| Space Complexity | $\mathcal{O}\left(\left(n\varepsilon^2/\log n\right)^{\frac{d}{2\alpha+2d}}\right)$ | $\mathcal{O}\left(\left(n\varepsilon^2/\log n\right)^{\frac{d}{2\alpha+2d}}\right)$ | $\mathcal{O}(nd)$ |

We compare the complexities of LPDT with other LDP regression methods in Table 1. Notably, [29] is inefficient due to its unacceptable test and space complexity. Also, the dominant term of training complexity of [9] is $\mathcal{O}(nd\log n\varepsilon^2)$. When $d$ is large, we can choose a small $n_q$ such that LPDT yields a strictly lower complexity than [9]. In addition, although [9] enjoys the same order of space complexity as LPDT, the memory of histogram-based methods suffers from the curse of dimensionality in practice. Since the storage of $\mathcal{O}(h^{-d})$ values is required, even $h = 1/2$ requires allocating an array of size $2^d$, which is problematic for large $d$. In contrast, LPDT can resist the curse of dimensionality by only splitting along the relevant features and keeping a small number of nodes.

## 4 Experiments

In the experiments, we first validate our theoretical findings with synthetic data in Section 4.1. Then, in Section 4.2, we show the superior performance of LPDT on real-world datasets with identically distributed public data. In Section 4.3, we apply LPDT to Chicago taxi data to show that LPDT

performs well even with considerable differences between private and public data. Also, we analyze the influence of the distribution shift between private and public data on LPDT.

**Splitting rule**    Note that most tree methods design their partition rules based on the information gained from the data. To boost the performance of LPDT, we also incorporate the variance reduction scheme from the original CART [11] to the tree construction. We denote the LPDT estimator using the max-edge partition rule with variance reduction in Section 2.3 as LPDT-M and denote the estimator using the standard variance reduction rule in [11] as LPDT-V. Since Theorem 3.1 holds for any partition, LPDT-V is also $\varepsilon$-LDP.

**Experiment setup**    We choose the privacy budget $\varepsilon \in [0.5, 8]$, covering commonly seen magnitudes of privacy budgets from low to high privacy regimes. We compare LPDT-M and LPDT-V with the following methods: (*i*) Private Histogram (PHIST) [9]. (*ii*) Adjusted Private Histogram (APHIST) [33]. (*iii*) Deconvolution Kernel (DECONV) [29]. Introduction to the methods and all implementation details are presented in Appendix D.1. We employ 5-fold cross-validation for parameter selection, and techniques for tuning parameters under LDP are discussed in Section D.2. The evaluation metric is the mean squared error (MSE).

## 4.1   Synthetic experiments

**Necessity of public data**    To demonstrate intuitively why public data is essential for LPDT, we first visualize its estimation on a synthetic model, $Y = \sin(16X) + \epsilon$ where $X \sim \mathcal{N}(0.5, 0.025)$ and $\varepsilon \sim \mathcal{N}(0, 1)$. In this case, the marginal distribution is highly imbalanced with the majority of samples located in the middle part of $[0, 1]$ and a few samples on the sides. For $\epsilon = 8$, we fit two LPDT models: one with 500 public data and 7,000 private data, and another with 8,000 private data.

As shown in Figure 1(a), without public data, LPDT struggles with the imbalanced marginal. The grids on the side produce unstable predictions since only a few samples fall into them. As a result, LPDT tends to decrease depth $s$ to stabilize its estimation. This leads to underfitting in the middle so that the predicted curve fails to capture the variation of the ground truth. In contrast, with the aid of public data, LPDT solves the issue as shown in Figure 1(b). For the mid-

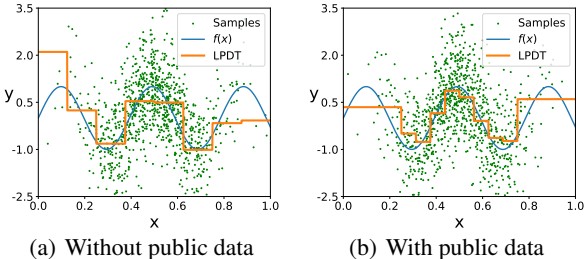

(a) Without public data          (b) With public data

Figure 1: The estimated regression curve of LPDT with and without public data. 1,000 samples are displayed in green.

dle zone where samples are abundant, LPDT creates small grids to enlarge approximation capacity. Meanwhile, it prunes the grids on the sides to ensure stability. Even with fewer private data, the MSE of LPDT is reduced from 1.19 to 1.08 thanks to the additional public data. In summary, the experiment provides empirical evidence supporting the theoretical findings in Theorems 3.4 and 3.5, which highlight the necessity of public data for the effective performance of LPDT.

**Parameter analysis of depth** $s$    We conduct experiments to investigate the selection of partition depth $s$ in terms of MSE. We generate 6,000 training samples, 2,000 test samples, and 2,000 public samples following the synthetic model described above. We pick $\varepsilon \in \{3, 4, 6, 8\}$ and $s \in \{2, 3, 4, 5, 6\}$. For each pair of $s$ and $\varepsilon$, we plot the 20 times averaged MSE versus $s$. The result is displayed in Figure 2(a). Apparently, for each $\varepsilon$, as $s$ increases, MSE first decreases until $s$ reaches a certain value. Then MSE begins to increase as $s$ grows. This further confirms the trade-off observed in Theorem 3.4. Moreover, the depth $s$ at which the test error is minimized increases as $\varepsilon$ increases. This is compatible with theory since the optimal choice of $s \asymp \log n\varepsilon^2$ is monotonically increasing with respect to $\varepsilon$.

**Parameter analysis of minimum leaf sample size** $n_l$    We conduct experiments to investigate the choice of $n_l$ in terms of MSE. Following the same generating scheme, we choose $\varepsilon \in \{3, 4, 6, 8\}$ and plot MSE of LPDT versus $n_l$ for $n_l \in \{15, 25, \cdots, 135\}$. In Figure 2(b), the relation between MSE and $n_l$ is U-shaped under each $\varepsilon$, which indicates that a properly chosen $n_l$ is necessary as stated in

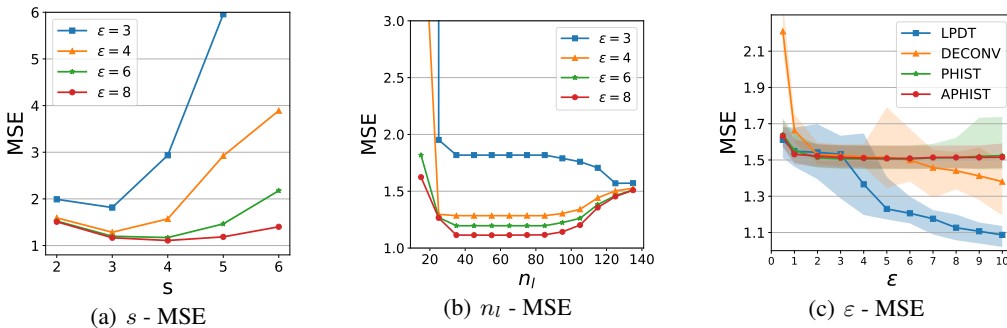

Figure 2: Different parameters versus MSE.

Theorem 3.4. Furthermore, LPDT achieves the best MSE for $n_l \in [35, 75]$ when $\varepsilon = 4, 6, 8$, while the minimum MSE occurred at $n_l = 125$ when $\varepsilon = 3$. This finding is compatible with Theorem 3.4 which states that the optimal choice of $n_l$ is monotonically decreasing with respect to $\varepsilon$.

Our parameter analyses indicate that decreasing $\varepsilon$ favors smaller values of $s$ and larger values of $n_l$. These choices lead to a decision tree partition with fewer grids. In summary, when facing higher levels of privacy demand, LPDT cuts down the number of grids to stabilize its estimation.

**Privacy utility trade-off**   We analyze how privacy budget $\varepsilon$ influences the quality of prediction. Under the same setup, we evaluate LPDT and other methods for $\varepsilon \in \{0.5, 1, 2, 3, \cdots, 10\}$ with 50 repetitions. The results are displayed in Figure 2(c). To show significance, we plot the $(0.1, 0.9)$ quantiles as confidence intervals. When $\varepsilon$ increases, the MSE of LPDT decreases much faster than the other methods. Note that the MSE of both PHIST and APHIST remains high, suggesting that their performances are limited by the histogram instead of the privacy mechanism.

## 4.2   Real data comparison with identically distributed public data

**Experiment setup**   We conduct experiments on 12 real datasets, each repeated 50 times with a ratio of 1:7:2 for public data, training data, and testing data in each trial. The dataset details and pre-processing steps are summarized in Appendix D.3. To ensure significance, we adopt the Wilcoxon signed-rank test [58] with a significance level of 0.05 to check if a result is significantly better. For better comparison, we also train a decision tree (denoted as DT) on the original training data with no privacy protection, whose result will serve as a lower bound.

Table 2: Average MSE over real data sets for LDP regression methods. The best results are **bolded** and the second best results are underlined. The marked results with significance towards the rest results are marked with ∗. Due to memory limitation, PHIST and APHIST are corrupted on two datasets which are marked with -.

| | DT | $\varepsilon = 2$ | | | | | $\varepsilon = 6$ | | | | |
|---|---|---|---|---|---|---|---|---|---|---|---|
| | | LPDT-M | LPDT-V | APHIST | PHIST | DECONV | LPDT-M | LPDT-V | APHIST | PHIST | DECONV |
| ABA | 5.67e+0 | **1.01e+1** | 1.01e+1 | 1.89e+1 | 1.06e+1 | 1.01e+7 | 8.38e+0* | **7.34e+0*** | 2.05e+1 | 1.05e+1 | 1.09e+1 |
| AIR | 2.26e+1 | 4.80e+1* | **4.69e+1*** | 1.31e+3 | 6.80e+1 | 3.00e+2 | 4.49e+1* | **3.60e+1*** | 1.60e+3 | 4.98e+1 | 4.72e+1 |
| ALG | 2.12e-2 | 2.57e-1 | **2.43e-1** | 2.52e-1 | 2.52e-1 | 9.26e+4 | **2.44e-1** | 2.46e-1 | 2.63e-1 | 2.47e-1 | 3.14e-1 |
| AQU | 1.92e+0 | 2.99e+0* | 2.99e+0* | 4.01e+0 | **2.93e+0*** | 5.74e+3 | 2.73e+0* | **2.67e+0*** | 4.75e+0 | 2.83e+0 | 2.96e+0 |
| BUI | 1.75e+5 | **1.50e+6*** | 1.64e+6* | - | - | 1.20e+9 | 1.44e+6* | **1.31e+6*** | - | - | 2.04e+7 |
| CBM | 4.08e-27 | 2.12e+0* | **1.65e+0*** | 9.53e+0 | 6.97e+0 | 2.37e+3 | 7.62e-1* | **1.23e-1*** | 4.94e+0 | 3.21e+0 | 1.23e+5 |
| CCP | 2.19e+1 | 1.50e+2* | **1.06e+2*** | 2.07e+4 | 3.64e+2 | 3.03e+2 | 8.42e+1* | **5.18e+1*** | 2.24e+4 | 3.28e+2 | 2.56e+2 |
| CON | 9.38e+1 | 2.94e+2* | **2.89e+2*** | 3.81e+2 | 3.00e+2 | 2.24e+7 | 2.44e+2* | **2.13e+2*** | 4.16e+2 | 2.96e+2 | 3.13e+2 |
| CPU | 2.15e+1 | 3.41e+2 | **9.00e+1*** | 9.26e+2 | 3.42e+2 | 2.15e+5 | 3.02e+2* | **6.15e+1*** | 9.98e+2 | 3.40e+2 | 3.98e+2 |
| FIS | 1.07e+0 | 2.15e+0* | **2.14e+0*** | 3.14e+0 | 2.22e+0 | 3.47e+3 | **1.65e+0*** | 1.76e+0* | 3.60e+0 | 2.16e+0 | 2.21e+0 |
| HOU | 2.11e+1 | **8.10e+1*** | 8.22e+1* | 1.06e+2 | 8.52e+1 | 1.92e+4 | 7.43e+1* | **7.10e+1*** | 1.23e+2 | 8.21e+1 | 2.44e+2 |
| MUS | 3.00e+2 | 3.47e+2* | **3.46e+2*** | - | - | 9.50e+3 | **3.27e+2*** | 3.27e+2* | - | - | 8.09e+3 |
| RED | 4.76e-1 | 7.08e-1* | **7.03e-1*** | 3.18e+0 | 7.57e-1 | 1.23e+8 | 6.75e-1* | **6.12e-1*** | 3.80e+0 | 7.12e-1 | 8.66e-1 |
| WHI | 5.77e-1 | 8.30e-1 | 8.42e-1 | 4.01e+0 | **8.15e-1** | 1.64e+7 | 7.03e-1* | **6.61e-1*** | 4.45e+0 | 8.03e-1 | 1.47e+0 |

**Performance of accuracy and running time**   The representative results for $\varepsilon = 2, 6$ are displayed in Table 2. Results of $\varepsilon = 0.5, 1, 4, 8$ is in Appendix D.4. It can be seen that LPDT-M and LPDT-V both significantly outperform their competitors. All methods achieve a higher MSE than DT, while the results for LPDT-M and LPDT-V are reasonably close to DT. Due to memory limitations, PHIST and APHIST fail on two datasets. We also compare the total running time in Table 6 in Appendix D.4. In general, both LPDT-M and LPDT-V achieve less running time than PHIST and APHIST, and are significantly faster than DECONV.

Table 3: Average MSE and standard deviation over Chicago taxi data.

| DT | | LPDT-M | | | | | | PHIST | | APHIST | |
|---|---|---|---|---|---|---|---|---|---|---|---|
| Public | Private | $\varepsilon = 0.5$ | $\varepsilon = 1$ | $\varepsilon = 2$ | $\varepsilon = 4$ | $\varepsilon = 6$ | $\varepsilon = 8$ | $\varepsilon = 2$ | $\varepsilon = 8$ | $\varepsilon = 2$ | $\varepsilon = 8$ |
| 3.71 | 0.80 | 113.45 (14.23) | 15.74 (2.20) | 4.89 (0.54) | 3.35 (0.33) | 2.86 (0.10) | 2.70 (0.10) | 24.72 (0.02) | 17.22 (0.00) | 38.22 (0.01) | 35.5 (0.01) |

With the private and public distributions being identical, another alternative is to use solely public data to fit a decision tree. We train such decision trees on public data ($n_q = 0.1n$). It is compared with the decision tree trained on private data and LPDT with $\varepsilon = 6$ in Table 7 in Appendix. The results show that when $n_q \ll n$ and $\varepsilon$ is large, training a decision tree on public data is worse than using LPDT on most of the datasets.

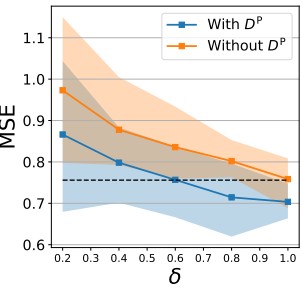

Figure 3: Power of public data.

**The power of public data**   We conduct experiments to show the utility gain brought by public data. We take red wine dataset as an example. With $n_q = 100$ public data, we run LPDT (with $D^P$) and PHIST (without $D^P$). For $\delta \in \{0.2, 0.4, 0.6, 0.8, 1\}$, we train each model on $\delta \cdot n$ samples with 20 repetitions. The result is in Figure 3. PHIST with 1,100 samples achieves the same MSE as LPDT with 660 samples. The result shows that, with a small amount of public data, LPDT achieves the same performance with much fewer samples.

### 4.3   Real data comparison with non-identically distributed public data

We apply LPDT to the *Chicago Taxi Dataset*, a collection of taxi trips in Chicago provided by the Differential Privacy Temporal Map Challenge and contains sensitive information [21]. We use the fare of each trip as labels and other information as features. Then we regard trips paid by PR card and credit card as public data and private data, respectively. In Appendix D.5, we show the two parts are distributed differently. After preprocessing, the dataset has 101 features with 2,150,565 samples in private data and 24,436 samples in public data.

**Performance**   We report the averaged MSE of LPDT over 20 repetitions for $\varepsilon = 0.5, 1, 2, 4, 6$, and 8. As a comparison to LPDT, we train two decision trees separately using the public and private datasets with no privacy protection. As for the comparison methods, DECONV fails due to the large sample size while PHIST and APHIST fail due to the dimensionality. However, after reducing the dimensionality by retaining only the continuous features, we are able to apply PHIST and APHIST. The results are displayed in Table 3. A first observation is that the decision tree trained on public data yields considerably worse results than the decision tree trained on private data. This suggests that relying solely on public data leads to biased predictions. Learning solely from public data achieves an MSE higher than LPDT for all three values of $\varepsilon$. Even for $\varepsilon = 2$, LPDT significantly outperforms histogram-based methods with $\varepsilon = 8$. This indicates that LPDT remains effective even when substantial disparities exist between the distributions of public and private data.

**How does non-identically distributed public data help?**   It is counterintuitive that non-identically distributed public data can benefit private estimation. We show the logic behind using such public data by investigating the first split feature in the tree partition. LPDT identifies whether a trip ends in Zone 32 as an important feature for predicting fare and initiates the recursive partitioning process by splitting along this feature. Figure 4 illustrates that trips ending in this zone generally have lower fares for both public and private data, although the actual fares differ significantly between the two datasets. This observation demonstrates that despite having different distributions, public and private data may exhibit similar patterns, thereby allowing the partition created on public data to still be effective on private data. Following this line of reasoning, to determine whether public data is suitable for a specific private task, we can examine whether the qualitative relationships between labels and features remain consistent across both datasets. Whether a dataset can be utilized as public data can be of independent interest as in [32].

**Analysis of distribution shift**   In Table 3, the MSE of LPDT reduces as $\varepsilon$ increases but remains higher than the MSE achieved by training a decision tree directly on private data. The performance

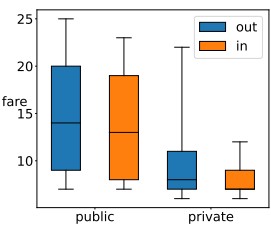
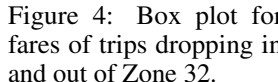
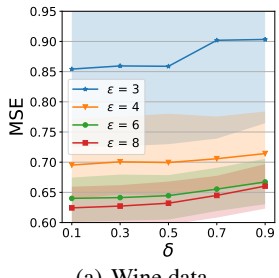
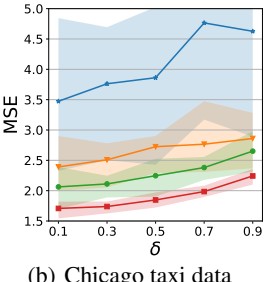

Figure 4: Box plot for fares of trips dropping in and out of Zone 32.

(a) Wine data

(b) Chicago taxi data

Figure 5: Portion of public data versus MSE of LPDT.

gap can be attributed to the distribution shift between private data and public data. In the following, we investigate how the difference between the two datasets affects the performance of LPDT. Besides the Chicago taxi data, we also adopt *White Wine* and *Red Wine* data in Section 4.2 as private and public data, respectively. The datasets contain the same variables but are distributed differently, as is investigated in transfer learning literature [48]. On both datasets, we combine part of the private samples with public data and perform the partition procedure on the combined dataset, with the portion of public data denoted as $\delta$. When $\delta$ is small, there is less difference between data used for partition and training. We report the average MSE of LPDT for $\delta \in \{0.1, 0.3, 0.5, 0.7, 0.9\}$ after 20 repetitions. Figure 5 shows that a small $\delta$ leads to a lower MSE for both datasets and all values of $\varepsilon$. Thus, LPDT is more powerful when the public and private data are distributed similarly, which also justifies the necessity of Assumption 3.3.

# 5 Conclusion

This paper addresses the challenge of effectively performing LDP regression given both public data and private data by introducing the locally private decision tree. Due to the novel idea of leveraging public data, LPDT is accurate, efficient, and interpretable. Theoretically, we establish the privacy guarantee and optimal convergence rate of LPDT. In experiments, we show the superior performance of LPDT regardless of the disparities between private and public data.

# 6 Limitations and broader impact

LPDT addresses the challenge of effectively performing LDP regression given both public data and private data. Future work may explore methodologies for incorporating public data in private learning using other models, such as linear models, and investigate the theoretical advantages of using public data in private learning. One limitation of this paper is that the measure of similarity between the public and private distribution considers only the marginal distribution. One may consider the similarities with respect to the regression function, potentially following transfer learning literature [13]. Moreover, in practice, public data may have a different distribution than private data, which is damaging to the learning process. The drawback yields the significance of public data selection or public data quality test such as [32, 60]. In addition, the theoretical analysis deal with $n_q \gtrsim n^{\frac{d}{2\alpha+2d}}$ and $n_q = 0$, while the intermediate zone for $n_q$ remains unexplored.

## Acknowledgment

The authors would like to thank the reviewers for their constructive comments, which led to a significant improvement in this work. The project is supported by the Special Funds of the National Natural Science Foundation of China (Grant No. 72342010) and was supported in part by the National Key R&D Program of China (2022YFB2702401). This research was supported by Public Computing Cloud, Renmin University of China.

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

# Appendix

In this appendix, we provide the detailed results for the additional methodology (Appendix A), the error analysis of the main theoretical results (Appendix B), the full proof of all theoretical results (Appendix C), and details as well as additional results of experiments (Appendix D).

## A Methodology of locally differentially private decision tree

### A.1 Algorithm of max-edge partition with variance reduction rule

---

**Algorithm 2:** Max-edge Partition Rule with Variance Reduction

---

**Input:** Public data $D^{pub}$, depth $s$, minimum leaf sample size $n_l$.
**Initialization:** $\pi_0 = [0,1]^d$.
**for** $i = 1$ **to** $s$ **do**
$\quad \pi_i = \emptyset$
$\quad$ **for** $j$ *in* $\mathcal{I}_{i-1}$ **do**
$\quad\quad$ Select $\ell$ as in (6).
$\quad\quad$ **if** $\sum_{i=1}^{n} \mathbf{1}\{X_i^{pub} \in A_{i-1}^{j,k}(\ell)\} \leq n_l$ **for** $k = 0, 1$ **then**
$\quad\quad\quad \pi_i = \pi_i \cup A_{i-1}^{j}$
$\quad\quad$ **end**
$\quad\quad$ **else**
$\quad\quad\quad \pi_i = \pi_i \cup \{A_{i-1}^{j,0}(\ell), A_{i-1}^{j,1}(\ell)\}$
$\quad\quad$ **end**
$\quad$ **end**
**end**
**Output:** Partition $\pi_s$

---

## B Error analysis

We rely on the following decomposition.

$$\mathcal{R}_{L,\mathrm{P}}(f_\pi^{\mathrm{DP}}) - \mathcal{R}_{L,\mathrm{P}}(f_\pi) = \underbrace{\mathcal{R}_{L,\mathrm{P}}(f_\pi^{\mathrm{DP}}) - \mathcal{R}_{L,\mathrm{P}}(f_\pi)}_{\textbf{Privatized Error}} + \underbrace{\mathcal{R}_{L,\mathrm{P}}(f_\pi) - \mathcal{R}_{L,\mathrm{P}}(\overline{f}_\pi)}_{\textbf{Sample Error}}$$

$$+ \underbrace{\mathcal{R}_{L,\mathrm{P}}(\overline{f}_\pi) - \mathcal{R}_{L,\mathrm{P}}(f^*)}_{\textbf{Approximation Error}}. \tag{7}$$

where $\overline{f}_\pi$, $f_\pi$ and $f_\pi^{\mathrm{DP}}$ are defined in (1), (2), and (5), respectively. Loosely speaking, the first error term quantifies the depravation brought by adding privacy noises to the estimator, which we call the privatized error. The second term corresponds to the expected estimation error brought by the randomness of the data, which we call the sample error. The last term is called approximation error, which arises due to the limited approximation capacity of piecewise constant functions. The following three lemmas provide bounds for each of the three errors.

**Lemma B.1** (**Bounding of Privatised Error**). *Let $f_\pi^{\mathrm{DP}}$ be the LPDT estimator in Algorithm 1. Suppose $f_\pi$ is the decision tree regressor defined in (2). Suppose $\pi$ is generated by Algorithm 2. Let Assumption 3.2 and 3.3 hold. Then, if we take $n_l \asymp n_q/2^s$, there holds*

$$\mathcal{R}_{L,\mathrm{P}}(f_\pi^{\mathrm{DP}}) - \mathcal{R}_{L,\mathrm{P}}(f_\pi) \lesssim \frac{2^{2s} \cdot \log n}{n\varepsilon^2}$$

*with probability $\mathrm{P}^n \otimes \mathrm{Q}^{n_q} \otimes \mathrm{R}^n$ at least $1 - 1/n_q^2 - 4/n^2$.*

**Lemma B.2** (**Bounding of Sample Error**). *Suppose $f_\pi$ and $\overline{f}_\pi$ are the decision tree regressor and the population decision tree regressor defined in (2) and (1), respectively. Let Assumption 3.2, and 3.3 hold. Then, if we take $n_l \asymp n_q/2^s$, there holds*

$$\mathcal{R}_{L,\mathrm{P}}(f_\pi) - \mathcal{R}_{L,\mathrm{P}}(\overline{f}_\pi) \lesssim \frac{2^s}{n}$$

*with probability $\mathrm{P}^n \otimes \mathrm{Q}^{n_q}$ at least $1 - 1/n_q^2 - 1/n^2$.*

**Lemma B.3** (**Bounding of Approximation Error**). *Suppose $\overline{f}_\pi$ is the population decision tree regressor defined in* (1). *Let Assumption 3.2 holds. Then there holds*

$$\mathcal{R}_{L,\mathrm{P}}(\overline{f}_\pi) - \mathcal{R}_{L,\mathrm{P}}(f^*) \leq 2c_L^2 2^{-(2\alpha s/d)\wedge s}.$$

## C  Proofs

### C.1  Proof of Theorem 3.1

***Proof of Theorem 3.1.*** For each conditional distribution $\mathrm{R}\left(\tilde{U}_i, \tilde{Y}_i | X_i = x, Y_i = y\right)$, we can compute the density ratio as

$$\sup_{x,x'\in\mathbb{R}^d,\ y,y'\in[-M,M],S\in\sigma(\mathcal{S})} \frac{\mathrm{R}\left((\tilde{U},\tilde{Y})\in S | X_i = x, Y_i = y\right)}{\mathrm{R}\left((\tilde{U},\tilde{Y})\in S | X_i = x', Y_i = y'\right)}$$

$$\leq \sup_{x,x'\in\mathbb{R}^d} \frac{d\mathrm{R}(\tilde{U}|X_i = x)}{d\mathrm{R}(\tilde{U}|X_i = x')} \cdot \sup_{y,y'\in[-M,M]} \frac{d\mathrm{R}(\tilde{Y}|Y_i = y)}{d\mathrm{R}(\tilde{Y}|Y_i = y')}. \tag{8}$$

(*i*) The first part is characterized by the random perturbation mechanism. Since the conditional density is identical if $x$ and $x'$ belongs to a same $A_j$, we have

$$\sup_{x,x'\in\mathbb{R}^d} \frac{d\mathrm{R}(\tilde{U}|X_i = x)}{d\mathrm{R}(\tilde{U}|X_i = x')} = \sup_{u,u'\in\{0,1\}^{|\mathcal{I}|}} \frac{d\mathrm{R}(\tilde{U}|U_i = u)}{d\mathrm{R}(\tilde{U}|U_i = u')}.$$

The right-hand side can be computed as

$$\frac{d\mathrm{R}(\tilde{U}|U_i = u)}{d\mathrm{R}(\tilde{U}|U_i = u')} = \frac{\prod_{j\in\mathcal{I}}\mathbb{P}[\tilde{U}_i^j|U_i^j = u^j]}{\prod_{j\in\mathcal{I}}\mathbb{P}[\tilde{U}_i^j|U_i^j = u'^j]}. \tag{9}$$

By definition, $u$ and $u'$ are one-hot vectors and differ at most on two entries. Without loss of generality, assume they differ on the first two elements. Also, the procedure in (3) yields that $e^{-\varepsilon/4} \leq \mathbb{P}[\tilde{U}_i^j|U_i^j = 0]/\mathbb{P}[\tilde{U}_i^j|U_i^j = 0] \leq e^{\varepsilon/4}$. As a result, (9) becomes

$$\sup_{u,u'\in\{0,1\}^{|\mathcal{I}|}} \frac{d\mathrm{R}(\tilde{U}|U_i = u)}{d\mathrm{R}(\tilde{U}|U_i = u')} = \sup_{u,u'\in\{0,1\}^{|\mathcal{I}|}} \frac{\prod_{j=1,2}\mathbb{P}[\tilde{U}_i^j|U_i^j = u^j]}{\prod_{j=1,2}\mathbb{P}[\tilde{U}_i^j|U_i^j = u'^j]} \leq e^{\varepsilon/4+\varepsilon/4} = e^{\varepsilon/2}. \tag{10}$$

(*ii*) For the second part, there holds

$$\sup_{y,y'\in[-M,M]} \frac{d\mathrm{R}(\tilde{Y}|Y_i = y)}{d\mathrm{R}(\tilde{Y}|Y_i = y')} = \sup_{y,y'\in[-M,M]} \frac{\exp\left(-\frac{\varepsilon}{4M}|\tilde{Y}-y|\right)}{\exp\left(-\frac{\varepsilon}{4M}|\tilde{Y}-y'|\right)}$$

$$\leq \sup_{y,y'\in[-M,M]} \exp\left(\frac{\varepsilon}{4M}|y-y'|\right) \leq e^{\varepsilon/2}. \tag{11}$$

Bringing (10) and (11) into (8) yields the desired conclusion. $\qquad\square$

### C.2  Proof of Theorem 3.4

***Proof of Theorem 3.4.*** Bringing Proposition B.1, B.2, and B.3 into the decomposition (7), we have

$$\mathcal{R}_{L,\mathrm{P}}(f_\pi^{\mathrm{DP}}) - \mathcal{R}_{L,\mathrm{P}}(f_\pi) \lesssim \frac{2^{2s}\cdot\log n}{n\varepsilon^2} + \frac{2^s}{n} + 2^{-(2\alpha s/d)\wedge s}$$

holds with probability $\mathrm{P}^n \otimes \mathrm{Q}^{n_q} \otimes \mathrm{R}^n$ at least $1 - 2/n_q^2 - 5/n^2$. When $(2\alpha s/d) \wedge p = 2\alpha s/d$, by choosing $2^s \asymp (n\varepsilon^2/\log n)^{d/(2\alpha+2d)}$, i.e. $s \asymp \log n\varepsilon^2$, we obtain $\mathcal{R}_{L,\mathrm{P}}(f_\pi^{\mathrm{DP}}) - \mathcal{R}_{L,\mathrm{P}}(f_\pi) \lesssim (n\varepsilon^2/\log n)^{-\frac{\alpha}{\alpha+d}}$. When $(2\alpha s/d) \wedge s = s$, by choosing $2^s \asymp (n\varepsilon^2/\log n)^{1/3}$, i.e. $s \asymp \log n\varepsilon^2$, we obtain $\mathcal{R}_{L,\mathrm{P}}(f_\pi^{\mathrm{DP}}) - \mathcal{R}_{L,\mathrm{P}}(f_\pi) \lesssim (n\varepsilon^2/\log n)^{-1/3}$. Note that $2\alpha s/d \geq s$ if and only if $\frac{\alpha}{\alpha+d} \geq 1/3$. As a result, we get

$$\mathcal{R}_{L,\mathrm{P}}(f_\pi^{\mathrm{DP}}) - \mathcal{R}_{L,\mathrm{P}}(f_\pi) \lesssim \left(\frac{\log n}{n\varepsilon^2}\right)^{\frac{\alpha}{\alpha+d}\wedge\frac{1}{3}}$$

with probability $\mathrm{P}^n \otimes \mathrm{Q}^{n_q} \otimes \mathrm{R}^n$ at least $1 - 2/n_q^2 - 5/n^2$. $\qquad\square$

## C.3 Proof of Theorem 3.5

*Proof of Theorem 3.5.* We consider such $P_c \in \mathcal{P}$ such that for each $dP_{c,X}(x) = c$ for some constant $c$ and $x \in [0, 1/2]$, and $dP_{c,Y|X}(0) = 1$. Then for such $P_c$, the Bayes function is $f(x) = 0$ and $\mathcal{R}^*_{L,P_c} = 0$. For the excess risk of $f_\pi^{DP}$, we have

$$\mathcal{R}_{L,P}(f_\pi^{DP}) = \sum_{j \in \mathcal{I}} \int_{A_j} \left( \frac{\sum_{i=1}^n \tilde{Y}_i \cdot \tilde{U}_i^j}{\sum_{i=1}^n \tilde{U}_i^j} \right)^2 dP_{c,X}(x)$$

$$= \sum_{j \in \mathcal{I}} \frac{16M^2}{\varepsilon^2} \left( \frac{\sum_{i=1}^n \xi_i \cdot \tilde{U}_i^j}{\sum_{i=1}^n \tilde{U}_i^j} \right)^2 \int_{A_j} dP_{c,X}(x). \tag{12}$$

We first take expectation with respect to $\xi$. There holds

$$\mathbb{E}_\xi \left[ \left( \sum_{i=1}^n \xi_i \cdot \tilde{U}_i^j \right)^2 \right] = \sum_{i=1}^n \mathbb{E}_\xi \xi_i^2 (\tilde{U}_i^j)^2. \tag{13}$$

For standard Laplace random variable $\xi$, there holds $\mathbb{E}_\xi \xi^2 = 2$. Each $\tilde{U}_i^j$ takes either $1 - \frac{1}{e^{\varepsilon/4}+1}$ or $-\frac{1}{e^{\varepsilon/4}+1}$, which implies $(\tilde{U}_i^j)^2 \geq \frac{1}{(e^{\varepsilon/4}+1)^2}$. Together, (13) becomes

$$\mathbb{E}_\xi \left[ \left( \sum_{i=1}^n \xi_i \cdot \tilde{U}_i^j \right)^2 \right] \geq \frac{2n}{(e^{\varepsilon/4}+1)^2}.$$

Bringing this into (12) leads to

$$\mathbb{E}_\xi \left[ \mathcal{R}_{L,P}(f_\pi^{DP}) \right] \geq \frac{32M^2 n}{\varepsilon^2 (e^{\varepsilon/4}+1)^2} \sum_{j \in \mathcal{I}} \left( \frac{1}{\sum_{i=1}^n \tilde{U}_i^j} \right)^2 \int_{A_j} dP_{c,X}(x). \tag{14}$$

Next, we take the expectation of $\tilde{U}_i^j$ with respect to both $P$ and $P_R$. Specifically, we have

$$\mathbb{E}_{\tilde{U}} \left[ \frac{1}{(\sum_{i=1}^n \tilde{U}_i^j)^2} \right] = \int_0^\infty \mathbb{P} \left( \frac{1}{(\sum_{i=1}^n \tilde{U}_i^j)^2} > t \right) dt$$

$$= \int_0^\infty \mathbb{P} \left( -\frac{1}{\sqrt{nt}} \leq \frac{1}{\sqrt{n}} \sum_{i=1}^n \tilde{U}_i^j \leq \frac{1}{\sqrt{nt}} \right) dt$$

$$\geq \int_0^{1/n} \mathbb{P} \left( -\frac{1}{\sqrt{nt}} \leq \frac{1}{\sqrt{n}} \sum_{i=1}^n \tilde{U}_i^j \leq \frac{1}{\sqrt{nt}} \right) dt \tag{15}$$

For any $n$, there exists some $P_c$ with $c < 1/(2\sqrt{n})$. For such $P_c$, we have

$$\sqrt{n} \int_{A_j} dP_{c,X}(x) = \sqrt{n} \cdot 2^{-s} \cdot c \leq 2^{-s-1}$$

since when $n_l = 0$, each grid has volumn $2^{-s}$. When $t < 1/n$, we have $1/\sqrt{nt} \geq 1 \geq 2 \cdot 2^{-s-1}$ for any $s$, which implies $1/\sqrt{nt} \geq 2 \cdot \sqrt{n} \int_{A_j} dP_{c,X}(x)$. Bring this into (15), we have

$$\mathbb{E}_{\tilde{U}} \left[ \frac{1}{(\sum_{i=1}^n \tilde{U}_i^j)^2} \right] \geq \int_0^{1/n} \mathbb{P} \left( -\frac{1}{2\sqrt{nt}} \leq \frac{1}{\sqrt{n}} \sum_{i=1}^n \tilde{U}_i^j - \sqrt{n} \int_{A_j} dP_{c,X}(x) \leq \frac{1}{2\sqrt{nt}} \right) dt. \tag{16}$$

Using the central limit theorem, we know

$$\mathbb{P} \left( -\frac{1}{2\sqrt{nt}} \leq \frac{1}{\sqrt{n}} \sum_{i=1}^n \tilde{U}_i^j - \sqrt{n} \int_{A_j} dP_{c,X}(x) \leq \frac{1}{2\sqrt{nt}} \right) \gtrsim 1$$

for $t \leq 1/n$. Then, (16) becomes

$$\mathbb{E}_{\tilde{U}} \left[ \frac{1}{(\sum_{i=1}^n \tilde{U}_i^j)^2} \right] \gtrsim \int_0^{1/n} dt = \frac{1}{n}.$$

This together with (14) yields

$$\mathbb{E} \left[ \mathcal{R}_{L,\mathrm{P}}(f_\pi^{\mathrm{DP}}) \right] \geq \frac{32M^2 n}{\varepsilon^2 (e^{\varepsilon/4} + 1)^2} \sum_{j \in \mathcal{I}} \mathbb{E}_{\tilde{U}} \left[ \left( \frac{1}{\sum_{i=1}^n \tilde{U}_i^j} \right)^2 \right] \int_{A_j} d\mathrm{P}_{c,X}(x)$$

$$\gtrsim n \sum_{j \in \mathcal{I}} \frac{1}{n} \int_{A_j} d\mathrm{P}_{c,X}(x) = \sum_{j \in \mathcal{I}} \int_{A_j} d\mathrm{P}_{c,X}(x) = 1.$$

$\square$

## C.4 Proofs of results in Section B

To prove lemmas in Section B, we first present several technical results. Their proof will be found in Section C.4

**Lemma C.1.** *Suppose $\zeta_i, i = 1, \cdots, n$ are independent random variables such that $a_i \leq \zeta_i \leq b_i$. Then there holds*

$$\mathbb{P} \left[ \left| \frac{1}{n} \sum_{i=1}^n \zeta_i - \mathbb{E} \frac{1}{n} \sum_{i=1}^n \zeta_i \right| \geq t \right] \leq 2e^{-\frac{2n^2 t^2}{\sum_{i=1}^n (b_i - a_i)^2}}$$

*for any $t > 0$.*

**Lemma C.2.** *Suppose $\xi_i, i = 1, \cdots, n$ are independent standard sub-exponential random variables with parameters $(\nu, \beta)$ [55, Definition 2.9]. Then there holds*

$$\mathbb{P} \left[ \left| \frac{1}{n} \sum_{i=1}^n \xi_i - \mathbb{E} \frac{1}{n} \sum_{i=1}^n \xi_i \right| \geq t \right] \leq 2e^{-\frac{nt^2}{2\nu^2}}.$$

*for any $t > 0$. Moreover, a standard Laplace random variable is sub-exponential with parameters $(\sqrt{2}, 1)$.*

**Lemma C.3.** *Let $\pi$ be a partition generated from Algorithm 2. Suppose Assumption 3.2 holds. Then for any $D = \{(X_1, Y_1), \cdots, (X_n, Y_n)\}$ drawn i.i.d. from $\mathrm{P}$ and any $x \in \mathcal{X}$, there holds*

$$\left| \frac{1}{n} \sum_{i=1}^n \mathbf{1}\{X_i \in A(x)\} - \int_{A(x)} d\mathrm{P}_X(x') \right| \leq \sqrt{\frac{\bar{c} \cdot 2^{1-s}(4d+5) \log n}{n}} + \frac{2(4d+5) \log n}{3n} + \frac{4}{n}$$

*with probability $\mathrm{P}^n$ at least $1 - 1/n^2$.*

**Lemma C.4.** *Let $\pi$ be a partition generated from Algorithm 2. Suppose Assumption 3.2 holds. Then for any $D = \{(X_1, Y_1), \cdots, (X_n, Y_n)\}$ drawn i.i.d. from $\mathrm{P}$ and any $x \in \mathcal{X}$, there holds*

$$\left| \frac{1}{n} \sum_{i=1}^n Y_i \mathbf{1}\{X_i \in A(x)\} - \int_{A(x)} f^*(x') d\mathrm{P}_X(x') \right|$$

$$\leq M \sqrt{\frac{\bar{c} \cdot 2^{1-s}(4d+5) \log n}{n}} + \frac{2M(4d+5) \log n}{3n} + \frac{4M}{n}$$

*with probability $\mathrm{P}^n$ at least $1 - 1/n^2$.*

**Lemma C.5.** *Let $\pi$ be a partition generated from Algorithm 2 using public data set $D^{pub} = \{(X_1^{pub}, Y_1^{pub}), \cdots, (X_{n_q}^{pub}, Y_{n_q}^{pub})\}$. Suppose Assumption 3.2 and 3.3 hold. Suppose $n \geq n_q \geq 2^s \log^2 n$. If we take $n_l = n_q/2^s$, there holds*

$$\int_{A(x)} d\mathrm{P}_X(x') \asymp 2^{-s}$$

*with probability* $1 - 1/n_q^2$. *Moreover, the analogous conclusion holds for the empirical probability measure, i.e.*

$$\frac{1}{n} \sum_{i=1}^{n} \mathbf{1}\{X_i \in A(x)\} \asymp 2^{-s}.$$

*with probability* $1 - 1/n_q^2 - 1/n^2$.

**Lemma C.6.** *Let* $\pi := \left\{A_i^j, i \in \{1, \cdots, p\}, j \in \{1, \cdots, 2^s\}\right\} = \{A_j, j \in \mathcal{I}\}$ *be the variance reduction decision tree partition generated by Algorithm 2. Then for any* $A_i^j := \times_{i=1}^{d}[a_i, b_i] \in \pi$, *denote the depth of* $A_i^j$ *as* $\operatorname{depth}(A_i^j) = i$. *Then for any* $A_i^j \in \pi$, *there holds*

$$2^{-1}\sqrt{d} \cdot 2^{-i/d} \le \operatorname{diam}(A_i^j) \le 2\sqrt{d} \cdot 2^{-i/d}$$

*This implies that for any* $A_j \in \pi$, *we have*

$$2^{-1}\sqrt{d} \cdot 2^{-\operatorname{depth}(A_j)/d} \le \operatorname{diam}(A_j) \le 2\sqrt{d} \cdot 2^{-\operatorname{depth}(A_j)/d}.$$

**Lemma C.7.** *Let* $\pi := \left\{A_i^j, i \in \{1, \cdots, p\}, j \in \{1, \cdots, 2^s\}\right\} = \{A_j, j \in \mathcal{I}\}$ *be the variance reduction decision tree partition generated by Algorithm 2. Then for any* $A_i^j \in \pi$, *denote the depth of* $A_i^j$ *as* $\operatorname{depth}(A_i^j) = i$. *Then there holds*

$$\sum_{j \in \mathcal{I}} 2^{p - \operatorname{depth}(A_j)} = 2^s.$$

***Proof of Lemma B.1.*** We intend to bound

$$\mathcal{R}_{L,\mathrm{P}}(f_\pi^{\mathrm{DP}}) - \mathcal{R}_{L,\mathrm{P}}(f_\pi) = \int_{\mathcal{X}} \left| f_\pi^{\mathrm{DP}}(x) - f_\pi(x) \right|^2 d\mathrm{P}_X(x)$$

$$= \int_{\mathcal{X}} \left| \frac{\sum_{i,j} \mathbf{1}\{x \in A_j\} \tilde{Y}_i \cdot \tilde{U}_i^j}{\sum_{i,j} \mathbf{1}\{x \in A_j\} \tilde{U}_i^j} - \frac{\sum_{i,j} \mathbf{1}\{x \in A_j\} Y_i \cdot U_i^j}{\sum_{i,j} \mathbf{1}\{x \in A_j\} U_i^j} \right|^2 d\mathrm{P}_X(x).$$

For each $j \in \mathcal{I}$ and any $x \in A_j$, the point-wise error can be decomposed as

$$\left| \frac{\sum_{i,j} \mathbf{1}\{x \in A_j\} \tilde{Y}_i \cdot \tilde{U}_i^j}{\sum_{i,j} \mathbf{1}\{x \in A_j\} \tilde{U}_i^j} - \frac{\sum_{i,j} \mathbf{1}\{x \in A_j\} Y_i \cdot U_i^j}{\sum_{i,j} \mathbf{1}\{x \in A_j\} U_i^j} \right|^2$$

$$\le 3 \cdot \left( \underbrace{\left| \frac{\sum_{i=1}^{n}(\tilde{Y}_i - Y_i)\tilde{U}_i^j}{\sum_{i=1}^{n} \tilde{U}_i^j} \right|^2}_{(I)} + \underbrace{\left| \frac{\frac{e^{\varepsilon/4}+1}{e^{\varepsilon/4}-1} \sum_{i=1}^{n} Y_i\tilde{U}_i^j \sum_{i=1}^{n} U_i^j - \sum_{i=1}^{n} Y_i U_i^j \sum_{i=1}^{n} U_i^j}{\frac{e^{\varepsilon/4}+1}{e^{\varepsilon/4}-1} \sum_{i=1}^{n} \tilde{U}_i^j \sum_{i=1}^{n} U_i^j} \right|^2}_{(II)} \right.$$

$$\left. + \underbrace{\left| \frac{\sum_{i=1}^{n} Y_i U_i^j \sum_{i=1}^{n} U_i^j - \frac{e^{\varepsilon/4}+1}{e^{\varepsilon/4}-1} \sum_{i=1}^{n} Y_i U_i^j \sum_{i=1}^{n} \tilde{U}_i^j}{\frac{e^{\varepsilon/4}+1}{e^{\varepsilon/4}-1} \sum_{i=1}^{n} \tilde{U}_i^j \sum_{i=1}^{n} U_i^j} \right|^2}_{(III)} \right)$$

using triangular inequality. We bound the three parts separately.

*(i)* For the numerator of $(I)$, note that $\tilde{Y}_i - Y_i = \frac{4M}{\varepsilon}\xi_i$ and $\tilde{U}_i^j \in [-1, 1]$, Lemma C.2 yields that, $(\tilde{Y}_i - Y_i)\tilde{U}_i^j$ are sub-exponential random variables with parameter $\left(\frac{\varepsilon}{2\sqrt{2}M}, 1\right)$ and consequently

$$\left| \frac{1}{n} \sum_{i=1}^{n}(\tilde{Y}_i - Y_i)\tilde{U}_i^j \right| \le \sqrt{\frac{128M^2 \cdot \log n}{n\varepsilon^2}} \tag{17}$$

with probability at least $1 - 1/n^2$. For the denominator, applying Lemma C.1 yields

$$\left| \frac{1}{n} \frac{e^{\varepsilon/4} + 1}{e^{\varepsilon/4} - 1} \sum_{i=1}^{n} \tilde{U}_i^j - \frac{1}{n} \sum_{i=1}^{n} U_i^j \right| \le \sqrt{\frac{\log n}{n}} \tag{18}$$

with probability at least $1 - 2/n^2$. Moreover, by Lemma C.5, we have

$$\frac{1}{n}\sum_{i=1}^{n} U_i^j = \sum_{i=1}^{n} \mathbf{1}\{X_i \in A_j\} \gtrsim \frac{1}{2^s}$$

holds with probability at least $1 - 1/n_q^2 - 1/n^2$. Then, we can guarantee that

$$\left|\frac{1}{n}\sum_{i=1}^{n}\tilde{U}_i^j\right| \geq \left|\frac{e^{\varepsilon/4}-1}{e^{\varepsilon/4}+1}\frac{1}{n}\sum_{i=1}^{n}U_i^j\right| - \left|\frac{1}{n}\sum_{i=1}^{n}\tilde{U}_i^j - \frac{e^{\varepsilon/4}-1}{e^{\varepsilon/4}+1}\frac{1}{n}\sum_{i=1}^{n}U_i^j\right| \tag{19}$$

$$\gtrsim \frac{1}{2^s} - \frac{e^{\varepsilon/4}-1}{e^{\varepsilon/4}+1}\sqrt{\frac{\log n}{n}} \gtrsim \frac{1}{2^s} \tag{20}$$

with probability $1 - 1/n_q^2 - 3/n^2$. This together with (17) yields

$$(I) \lesssim \frac{2^{2s} \cdot \log n}{n\varepsilon^2}. \tag{21}$$

(*ii*) For $(II)$, we first cancel the $\sum_{i=1}^{n} U_i^j$ on both the numerator and the denominator. Then, applying Lemma C.1, we get

$$\left|\frac{1}{n}\frac{e^{\varepsilon/4}+1}{e^{\varepsilon/4}-1}\sum_{i=1}^{n}Y_i\tilde{U}_i^j - \frac{1}{n}\sum_{i=1}^{n}Y_iU_i^j\right| \leq \sqrt{\frac{M \cdot \log n}{n}}$$

with probability $1 - 2/n^2$ since $|Y_i| \leq M$. Combining this with (19), we get

$$(II) \lesssim \frac{2^{2s} \cdot \log n}{n}. \tag{22}$$

(*iii*) For $(III)$, note that $U_i^j \in \{0,1\}$, we have $\sum_{i=1}^{n} Y_iU_i^j \leq M\sum_{i=1}^{n} U_i^j$. Thus,

$$\left|\frac{\sum_{i=1}^{n}Y_iU_i^j\sum_{i=1}^{n}U_i^j - \frac{e^{\varepsilon/4}+1}{e^{\varepsilon/4}-1}\sum_{i=1}^{n}Y_iU_i^j\sum_{i=1}^{n}\tilde{U}_i^j}{\frac{e^{\varepsilon/4}+1}{e^{\varepsilon/4}-1}\sum_{i=1}^{n}\tilde{U}_i^j\sum_{i=1}^{n}U_i^j}\right|^2 \leq M^2\left|\frac{\sum_{i=1}^{n}U_i^j - \frac{e^{\varepsilon/4}+1}{e^{\varepsilon/4}-1}\sum_{i=1}^{n}\tilde{U}_i^j}{\frac{e^{\varepsilon/4}+1}{e^{\varepsilon/4}-1}\sum_{i=1}^{n}\tilde{U}_i^j}\right|^2$$

$$\lesssim M^2\frac{2^{2s} \cdot \log n}{n}$$

where the last inequality follows from (18) and (19). Combining this with (21) and (22), we have

$$\left|\frac{\sum_{i,j}\mathbf{1}\{x \in A_j\}\tilde{Y}_i \cdot \tilde{U}_i^j}{\sum_{i,j}\mathbf{1}\{x \in A_j\}\tilde{U}_i^j} - \frac{\sum_{i,j}\mathbf{1}\{x \in A_j\}Y_i \cdot U_i^j}{\sum_{i,j}\mathbf{1}\{x \in A_j\}U_i^j}\right|^2 \lesssim \frac{2^{2s} \cdot \log n}{n\varepsilon^2}$$

with probability $\mathrm{P}^n \times \mathrm{Q}^{n_q} \times \mathrm{R}$ at least $1 - 4/n^2 - 1/n_q^2$ for any $x \in \mathcal{X}$. This directly implies the desired bound of $\mathcal{R}_{L,\mathrm{P}}(f_\pi^{\mathrm{DP}}) - \mathcal{R}_{L,\mathrm{P}}(f_\pi)$. $\square$

***Proof of Lemma B.2***. We intend to bound

$$\mathcal{R}_{L,\mathrm{P}}(f_\pi) - \mathcal{R}_{L,\mathrm{P}}(\overline{f}_\pi) = \int_{\mathcal{X}}\left|f_\pi^{\mathrm{DP}}(x) - f_\pi(x)\right|^2 d\mathrm{P}_X(x)$$

$$= \int_{\mathcal{X}}\left|\frac{\sum_{i,j}\mathbf{1}\{x \in A_j\}Y_iU_i^j}{\sum_{i,j}\mathbf{1}\{x \in A_j\}U_i^j} - \frac{\sum_{j\in\mathcal{I}_p}\mathbf{1}\{x \in A_j\}\int_{A_j}f^*(x')d\mathrm{P}_X(x)}{\sum_{j\in\mathcal{I}_p}\mathbf{1}\{x \in A_j\}\int_{A_j}d\mathrm{P}_X(x)}\right|^2 d\mathrm{P}_X(x).$$

For any fixed $j$, we have

$$\left| f_\pi^{\mathrm{DP}}(x) - f_\pi(x) \right|^2 \leq \left| \frac{\frac{1}{n}\sum_{i=1}^n Y_i U_i^j}{\frac{1}{n}\sum_{i=1}^n U_i^j} - \frac{\int_{A_j} f^*(x')dP_X(x')}{\int_{A_j} dP_X(x)} \right|^2$$

$$\leq \underbrace{\left| \frac{\frac{1}{n}\sum_{i=1}^n Y_i U_i^j \int_{A_j} dP_X(x') - \int_{A_j} f^*(x')dP(x')\int_{A_j} dP_X(x')}{\frac{1}{n}\sum_{i=1}^n U_i^j \int_{A_j} dP_X(x)} \right|^2}_{(I)}$$

$$+ \underbrace{\left| \frac{\int_{A_j} f^*(x')dP(x')\int_{A_j} dP_X(x') - \frac{1}{n}\sum_{i=1}^n U_i^j \int_{A_j} f^*(x')dP_X(x')}{\frac{1}{n}\sum_{i=1}^n U_i^j \int_{A_j} dP_X(x)} \right|^2}_{(II)}$$

We bound two terms separately. For $(I)$, Lemma C.5 yields

$$\frac{1}{n}\sum_{i=1}^n U_i^j \int_{A_j} dP_X(x) \gtrsim 2^{-2s} \tag{23}$$

with probability $1 - 1/n_q^2 - 1/n^2$. For the numerator, Lemma C.4 yields

$$\left| \sum_{i=1}^n Y_i U_i^j - \int_{A_j} f^*(x')dP_X(x') \right|^2 \left| \int_{A_j} dP_X(x') \right|^2 \lesssim \frac{1}{2^s \cdot n} \cdot 2^{-2s}.$$

Together, we get $(I) \lesssim 2^s/n$. Analogously, by Lemma C.3, we have

$$\left| \int_{A_j} dP_X(x') - \sum_{i=1}^n U_i^j \right|^2 \left| \int_{A_j} f^*(x')dP_X(x') \right|^2 \leq \frac{1}{2^p \cdot n} \cdot M^2 \cdot 2^{-2s}.$$

This together with (23) yields $(II) \lesssim 2^s/n$. The bound of $(I)$ and $(II)$ together yields the desired conclusion. $\qquad\square$

***Proof of Lemma B.3***. We intend to bound

$$\mathcal{R}_{L,\mathrm{P}}(\overline{f}_\pi) - \mathcal{R}_{L,\mathrm{P}}(f^*) = \int_{x\in\mathcal{X}} \left( \frac{\int_{A(x)} f(x')dP_X(x')}{\int_{A(x)} dP_X(x')} - f(x) \right)^2 dP_X(x).$$

For each $x$, if Assumption 3.2 holds, the point-wise error can be bounded by

$$\frac{\int_{A(x)} f(x')dP_X(x')}{\int_{A(x)} dP_X(x')} - f(x) \leq \frac{\int_{A(x)} |f(x') - f(x)|dP_X(x')}{\int_{A(x)} dP_X(x')}$$

$$\leq \frac{\int_{A(x)} c_L \|x' - x\|^\alpha dP_X(x')}{\int_{A(x)} dP_X(x')} \leq c_L \mathrm{diam}(A(x))^\alpha$$

Then we can bound the integral of point-wise error by the sum of the integral of errors on each $A_j$, namely

$$\int_{x\in\mathcal{X}} \left( \frac{\int_{A(x)} f(x')dP_X(x')}{\int_{A(x)} dP_X(x')} - f(x) \right)^2 dP(x) \leq \sum_{j\in\mathcal{I}} c_L^2 \mathrm{diam}(A_j)^{2\alpha} \int_{A_j} dP_X(x). \tag{24}$$

Apply Lemma C.5 and C.6, we have

$$\sum_{j\in\mathcal{I}} c_L^2 \mathrm{diam}(A_j)^{2\alpha} \int_{A_j} dP_X(x) \leq \sum_{j\in\mathcal{I}} 2c_L^2 \sqrt{d} 2^{-2\alpha\cdot\mathrm{depth}(A_j)/d-s}$$

$$= \sum_{j\in\mathcal{I}} 2c_L^2 \sqrt{d} 2^{s-\mathrm{depth}(A_j)} \cdot 2^{-2\alpha\cdot\mathrm{depth}(A_j)/d+\mathrm{depth}(A_j)-2s}. \tag{25}$$

For each $j$, we can guarantee

$$2^{-2\alpha\cdot\text{depth}(A_j)/d+\text{depth}(A_j)} \leq 2^{(1-(2\alpha/d)\wedge 1)\text{depth}(A_j)} \leq 2^{(1-(2\alpha/d)\wedge 1)s} \qquad (26)$$

since $(2\alpha/d)\wedge 1 \leq 1$. Bringing (26) into (25), we get

$$\sum_{j\in\mathcal{I}} c_L^2\text{diam}(A_j)^{2\alpha}\int_{A_j} d\text{P}(x) \leq 2^{-s-(2\alpha s/d)\wedge s}\sum_{j\in\mathcal{I}} 2c_L^2\sqrt{d}2^{s-\text{depth}(A_j)}$$

By applying Lemma C.7, we immediately have

$$\sum_{j\in\mathcal{I}} c_L^2\text{diam}(A_j)^{2\alpha}\int_{A_j} d\text{P}(x) \leq 2c_L^2 2^{-(2\alpha s/d)\wedge s}.$$

This together with (24) yields the desired conclusion. □

## C.5 Proofs of results in Section C.4

***Proof of Lemma C.1***. The conclusion follows from Example 2.4 and Proposition 2.5 in [55]. □

***Proof of Lemma C.2***. The conclusion follows from (2.18) in [55]. □

In the subsequent proof, we define the empirical measure $\text{D} := \frac{1}{n}\sum_{i=1}^{n}\delta_{(X_i,Y_i)}$ given samples $D = \{(X_1, Y_1), \cdots, (X_n, Y_n)\}$, where $\delta$ is the Dirac function. To conduct our analysis, we first need to recall the definitions of *VC dimension* (*VC index*) and *covering number*, which are frequently used in capacity-involved arguments and measure the complexity of the underlying function class [54, 40, 31].

**Definition C.8** (VC dimension). Let $\mathcal{B}$ be a class of subsets of $\mathcal{X}$ and $A \subset \mathcal{X}$ be a finite set. The trace of $\mathcal{B}$ on $A$ is defined by $\{B \cap A : B \subset \mathcal{B}\}$. Its cardinality is denoted by $\Delta^{\mathcal{B}}(A)$. We say that $\mathcal{B}$ shatters $A$ if $\Delta^{\mathcal{B}}(A) = 2^{\#(A)}$, that is, if for every $A' \subset A$, there exists a $B \subset \mathcal{B}$ such that $A' = B \cap A$. For $n \in \text{N}$, let

$$m^{\mathcal{B}}(n) := \sup_{A\subset\mathcal{X},\,\#(A)=n}\Delta^{\mathcal{B}}(A). \qquad (27)$$

Then, the set $\mathcal{B}$ is a Vapnik-Chervonenkis class if there exists $n < \infty$ such that $m^{\mathcal{B}}(n) < 2^n$ and the minimal of such $n$ is called the VC dimension of $\mathcal{B}$, and abbreviate as $\text{VC}(\mathcal{B})$.

Since an arbitrary set of $n$ points $\{x_1, \ldots, x_n\}$ possess $2^n$ subsets, we say that $\mathcal{B}$ *picks out* a certain subset from $\{x_1, \ldots, x_n\}$ if this can be formed as a set of the form $B \cap \{x_1, \ldots, x_n\}$ for a $B \in \mathcal{B}$. The collection $\mathcal{B}$ *shatters* $\{x_1, \ldots, x_n\}$ if each of its $2^n$ subsets can be picked out in this manner. From Definition C.8 we see that the VC dimension of the class $\mathcal{B}$ is the smallest $n$ for which no set of size $n$ is shattered by $\mathcal{B}$, that is,

$$\text{VC}(\mathcal{B}) = \inf\Big\{n : \max_{x_1,\ldots,x_n}\Delta^{\mathcal{B}}(\{x_1,\ldots,x_n\}) \leq 2^n\Big\},$$

where $\Delta^{\mathcal{B}}(\{x_1, \ldots, x_n\}) = \#\{B \cap \{x_1, \ldots, x_n\} : B \in \mathcal{B}\}$. Clearly, the more refined $\mathcal{B}$ is, the larger is its index.

To further bound the capacity of the function sets, we need to introduce the following fundamental descriptions of *covering number* which enables an approximation of an infinite set by finite subsets.

**Definition C.9** (Covering Number). Let $(\mathcal{X}, d)$ be a metric space and $A \subset \mathcal{X}$. For $\varepsilon > 0$, the $\varepsilon$-covering number of $A$ is denoted as

$$\mathcal{N}(A, d, \varepsilon) := \min\Big\{n \geq 1 : \exists x_1, \ldots, x_n \in \mathcal{X} \text{ such that } A \subset \bigcup_{i=1}^{n} B(x_i, \varepsilon)\Big\},$$

where $B(x, \varepsilon) := \{x' \in \mathcal{X} : d(x, x') \leq \varepsilon\}$.

To prove Lemma C.10, we need the following fundamental lemma concerning with the VC dimension of random partitions in Section 2.3, which follows the idea put forward by [30] of the construction of random forest. To this end, let $p \in \mathbb{N}$ be fixed and $\pi_p$ be a partition of $\mathcal{X}$ with number of splits $s$ and $\pi_{(p)}$ denote the collection of all partitions $\pi_p$.

**Lemma C.10.** *Let $\tilde{\mathcal{A}}$ be the collection of all cells $\times_{i=1}^{d}[a_i, b_i]$ in $\mathbb{R}^d$. The VC index of $\tilde{\mathcal{A}}$ equals $2d+1$. Moreover, for all $0 < \varepsilon < 1$, there exists a universal constant $C$ such that*

$$\mathcal{N}(\mathbf{1}_{\tilde{\mathcal{A}}}, \|\cdot\|_{L_1(Q)}, \varepsilon) \leq C(2d+1)(4e)^{2d+1}(1/\varepsilon)^{2d}.$$

*Proof of Lemma C.10.* The first result of VC index follows from Example 2.6.1 in [54]. The second result of covering number follows directly from Theorem 9.2 in [40]. $\square$

Before we proceed, we list the well-known Bernstein's inequality that will be used frequently in the proofs. Lemma C.11 was introduced in [8] and can be found in many statistical learning textbooks, see e.g., [42, 19, 51].

**Lemma C.11** (Bernstein's inequality). *Let $B > 0$ and $\sigma > 0$ be real numbers, and $n \geq 1$ be an integer. Furthermore, let $\xi_1, \ldots, \xi_n$ be independent random variables satisfying $\mathbb{E}_P \xi_i = 0$, $\|\xi_i\|_\infty \leq B$, and $\mathbb{E}_P \xi_i^2 \leq \sigma^2$ for all $i = 1, \ldots, n$. Then for all $\tau > 0$, we have*

$$P\left(\frac{1}{n}\sum_{i=1}^{n}\xi_i \geq \sqrt{\frac{2\sigma^2\tau}{n}} + \frac{2B\tau}{3n}\right) \leq e^{-\tau}.$$

*Proof of Lemma C.3.* Let $\tilde{\mathcal{A}}$ be the collection of all cells $\times_{i=1}^{d}[a_i, b_i]$ in $\mathbb{R}^d$. Applying Lemma C.10 with $Q := (D_X + P_X)/2$, there exists an $\varepsilon$-net $\{\tilde{A}_k\}_{k=1}^{K} \subset \tilde{\mathcal{A}}$ with

$$K \leq C(2d+1)(4e)^{2d+1}(1/\varepsilon)^{2d} \tag{28}$$

such that for any $j \in \mathcal{I}_p$, there exist some $k \in \{1, \ldots, K\}$ such that

$$\|\mathbf{1}\{x \in A_p^j\} - \mathbf{1}\{x \in \tilde{A}_k\}\|_{L_1((D_X + P_X)/2)} \leq \varepsilon,$$

Since

$$\|\mathbf{1}\{x \in A_p^j\} - \mathbf{1}\{x \in \tilde{A}_k\}\|_{L_1((D_X + P_X)/2)}$$
$$= 1/2 \cdot \|\mathbf{1}\{x \in A_p^j\} - \mathbf{1}\{x \in \tilde{A}_k\}\|_{L_1(D_X)} + 1/2 \cdot \|\mathbf{1}\{x \in A_p^j\} - \mathbf{1}\{x \in \tilde{A}_k\}\|_{L_1(P_X)},$$

we get

$$\|\mathbf{1}\{x \in A_p^j\} - \mathbf{1}\{x \in \tilde{A}_k\}\|_{L_1(D_X)} \leq 2\varepsilon, \quad \|\mathbf{1}\{x \in A_p^j\} - \mathbf{1}\{x \in \tilde{A}_k\}\|_{L_1(P_X)} \leq 2\varepsilon. \tag{29}$$

Consequently, by the definition of the covering number and the triangle inequality, for any $j \in \mathcal{I}_p$, there holds

$$\left|\frac{1}{n}\sum_{i=1}^{n}\mathbf{1}\{x \in A_p^j\}(X_i) - \int_{\tilde{A}_p^j} dP_X(x')\right|$$

$$\leq \left|\frac{1}{n}\sum_{i=1}^{n}\mathbf{1}\{x \in \tilde{A}_k\}(X_i) - \int_{\tilde{A}_k} dP_X(x')\right| + \|\mathbf{1}\{x \in A_p^j\} - \mathbf{1}\{x \in \tilde{A}_k\}\|_{L_1(D_X)}$$

$$+ \|\mathbf{1}\{x \in A_p^j\} - \mathbf{1}\{x \in \tilde{A}_k\}\|_{L_1(P_X)} \leq \left|\frac{1}{n}\sum_{i=1}^{n}\mathbf{1}\{x \in \tilde{A}_k\}(X_i) - \int_{\tilde{A}_k} dP_X(x')\right| + 4\varepsilon.$$

Therefore, we get

$$\sup_{j \in \mathcal{I}}\left|\frac{1}{n}\sum_{i=1}^{n}\mathbf{1}\{x \in A_p^j\}(X_i) - \int_{\tilde{A}_p^j} dP_X(x')\right| \leq \sup_{1 \leq k \leq K}\left|\frac{1}{n}\sum_{i=1}^{n}\mathbf{1}\{x \in \tilde{A}_k\}(X_i) - \int_{\tilde{A}_k} dP_X(x')\right| + 4\varepsilon. \tag{30}$$

For any fixed $1 \leq k \leq K$, let the random variable $\xi_i$ be defined by $\xi_i := \mathbf{1}\{X_i \in \tilde{A}_k\} - \int_{\tilde{A}_k} dP_X(x')$. Then we have $\mathbb{E}_{P_X}\xi_i = 0$, $\|\xi\|_\infty \leq 1$, and $\mathbb{E}_{P_X}\xi_i^2 \leq \int_{\tilde{A}_k} dP_X(x')$. According to Assumption 3.2, there holds $\mathbb{E}_{P_X}\xi_i^2 \leq \bar{c} \cdot 2^{-s}$. Applying Bernstein's inequality in Lemma C.11, we obtain

$$\left|\frac{1}{n}\sum_{i=1}^{n}\mathbf{1}\{X_i \in \tilde{A}_k\} - \int_{\tilde{A}_k} dP_X(x')\right| \leq \sqrt{\frac{\bar{c} \cdot 2^{1-s} \cdot \tau}{n}} + \frac{2\tau \log n}{3n}$$

with probability $\mathrm{P}^n$ at least $1 - 2e^{-\tau}$. Then the union bound together with the covering number estimate (28) implies that

$$\sup_{1 \leq k \leq K} \left| \frac{1}{n} \sum_{i=1}^n \mathbf{1}\{X_i \in \tilde{A}_k\} - \int_{\tilde{A}_k} d\mathrm{P}_X(x') \right| \leq \sqrt{\frac{\overline{c} \cdot 2^{1-s}(\tau + \log(2K))}{n}} + \frac{2(\tau + \log(2K))\log n}{3n}$$

with probability $\mathrm{P}^n$ at least $1 - e^{-\tau}$. Let $\tau = 2\log n$ and $\varepsilon = 1/n$. Then for any $n > N_1 := (2C) \wedge (2d+1) \wedge (4e)$, we have $\tau + \log(2K) = 2\log n + \log(2C) + \log(2d+1) + (2d+1)\log(4e) + 2d\log n \leq (4d+5)\log n$. Therefore, we have

$$\sup_{1 \leq k \leq K} \left| \frac{1}{n} \sum_{i=1}^n \mathbf{1}\{X_i \in \tilde{A}_k\} - \int_{\tilde{A}_k} d\mathrm{P}_X(x') \right| \leq \sqrt{\frac{\overline{c} \cdot 2^{1-s}(4d+5)\log n}{n}} + \frac{2(4d+5)\log n}{3n} \tag{31}$$

with probability $\mathrm{P}^n$ at least $1 - 1/n^2$. This together with (30) yields that

$$\sup_{j \in \mathcal{I}} \left| \frac{1}{n} \sum_{i=1}^n \mathbf{1}\{x \in A_p^j\} - \int_{\tilde{A}_p^j} d\mathrm{P}_X(x') \right| \leq \sqrt{\frac{\overline{c} \cdot 2^{1-s}(4d+5)\log n}{n}} + \frac{2(4d+5)\log n}{3n} + \frac{4}{n}.$$

$\square$

***Proof of Lemma C.4.*** Let $\tilde{\mathcal{A}}$ be the collection of all cells $\times_{i=1}^d [a_i, b_i]$ in $\mathbb{R}^d$. Then there exists an $\varepsilon$-net $\{\tilde{A}_k\}_{k=1}^K \subset \tilde{\mathcal{A}}$ with $K$ bounded by (28) such that for any $j \in \mathcal{I}$, (29) holds for some $k \in \{1, \ldots, K\}$. Consequently, by the definition of the covering number and the triangle inequality, for any $j \in \mathcal{I}_p$, there holds

$$\left| \sum_{i=1}^n \mathbf{1}\{X_i \in A_p^j\}Y_i - \int_{A_p^j} f^*(x')d\mathrm{P}_X(x') \right|$$

$$\leq \left| \sum_{i=1}^n \mathbf{1}\{X_i \in \tilde{A}_k\}Y_i - \int_{\tilde{A}_k} f^*(x')d\mathrm{P}_X(x') \right|$$

$$+ \int_{\mathbb{R}^d} \left| \mathbf{1}\{x' \in A_p^j\} - \mathbf{1}\{x' \in \tilde{A}_k\} \right| |f^*(x')| d\mathrm{P}_X(x') + \sum_{i=1}^n \left| \mathbf{1}\{X_i \in \tilde{A}_k\} - \mathbf{1}\{X_i \in A_p^j\} \right| |Y_i|$$

$$\leq \left| \sum_{i=1}^n \mathbf{1}\{X_i \in \tilde{A}_k\}Y_i - \int_{\tilde{A}_k} f^*(x')d\mathrm{P}_X(x') \right|$$

$$+ \max_{1 \leq i \leq n} |Y_i| \cdot \|\mathbf{1}\{x \in A_p^j\} - \mathbf{1}\{x \in \tilde{A}_k\}\|_{L_1(\mathrm{D}_X)} + M \cdot \|\mathbf{1}\{x \in A_p^j\} - \mathbf{1}\{x \in \tilde{A}_k\}\|_{L_1(\mathrm{P}_X)}$$

$$\leq \left| \sum_{i=1}^n \mathbf{1}_{\tilde{A}_k}(X_i)Y_i - \int_{\tilde{A}_k} f^*(x')d\mathrm{P}_X(x') \right| + 4M\varepsilon. \tag{32}$$

where the last inequality follow from the condition $\mathcal{Y} \subset [-M, M]$.

For any fixed $1 \leq k \leq K$, let the random variable $\tilde{\xi}_i$ be defined by $\tilde{\xi}_i := \mathbf{1}\{X_i \in \tilde{A}_k\}Y_i - \int_{\tilde{A}_k} f^*(x') d\mathrm{P}_X(x')$. Then we have $\mathbb{E}_\mathrm{P}\tilde{\xi}_i = 0$, $\|\xi\|_\infty \leq 1$, and $\mathbb{E}_\mathrm{P}\tilde{\xi}_i^2 \leq M^2 \int_{\tilde{A}_k} d\mathrm{P}(x')$. According to Assumption 3.2, there holds $\mathbb{E}_\mathrm{P}\tilde{\xi}_i^2 \leq M^2 \cdot \overline{c} \cdot 2^{-s}$. Applying Bernstein's inequality in Lemma C.11, we obtain

$$\left| \sum_{i=1}^n \mathbf{1}\{X_i \in \tilde{A}_k\}Y_i - \int_{\tilde{A}_k} f^*(x')d\mathrm{P}_X(x') \right| \leq \sqrt{\frac{M^2 \cdot \overline{c} \cdot 2^{1-s} \cdot \tau}{n}} + \frac{2M\tau\log n}{3n}$$

with probability $\mathrm{P}^n$ at least $1 - 2e^{-\tau}$. Similar to the proof of Lemma C.3, one can show that for any $n \geq N_1$, there holds

$$\sup_{1 \leq k \leq K} \left| \sum_{i=1}^n \mathbf{1}\{X_i \in \tilde{A}_k\}Y_i - \int_{\tilde{A}_k} f^*(x')d\mathrm{P}_X(x') \right| \leq M\sqrt{\frac{\overline{c} \cdot 2^{1-s} \cdot \tau}{n}} + \frac{2M\tau\log n}{3n}$$

with probability $\mathrm{P}^n$ at least $1 - 1/n^2$. This together with (32) yields that

$$\left| \sum_{i=1}^{n} \mathbf{1}\{X_i \in A_p^j\} Y_i - \int_{A_p^j} f^*(x') d\mathrm{P}_X(x') \right| \tag{33}$$

$$\leq M\sqrt{\frac{\overline{c} \cdot 2^{1-s}(4d+5)\log n}{n}} + \frac{2M(4d+5)\log n}{3n} + \frac{4M}{n}. \tag{34}$$

$\square$

***Proof of Lemma C.5.*** Since $\pi$ is generated by Algorithm 2, the number of public samples in each of its leaf cell $A_j, j \in \mathcal{I}_p$ is no fewer than $n_l$, i.e.

$$\sum_{i=1}^{n} \mathbf{1}\{X_i^{pub} \in A_j\} \geq n_l, \ \ j \in \mathcal{I}_p$$

which implies $n_q^{-1} \sum_{i=1}^{n} \mathbf{1}\{X_i^{pub} \in A_j\} \geq 2^{-s}$ Hence, by Lemma C.3, there holds

$$\int_{A_j} d\mathrm{Q}_X(x') \geq \frac{1}{n_q} \sum_{i=1}^{n} \mathbf{1}\{X_i^{pub} \in A_j\} - \sqrt{\frac{\overline{c} \cdot 2^{1-s}(4d+5)\log n_q}{n_q}} - \frac{2(4d+5)\log n_q}{3n_q} - \frac{4}{n_q}$$

$$\geq \frac{1}{2^s} - \sqrt{\frac{\overline{c} \cdot 2^{1-s}(4d+5)\log n_q}{n_q}} - \frac{2(4d+5)\log n_q}{3n_q} - \frac{4}{n_q}$$

with probability $\mathrm{Q}^{n_q}$ at least $1 - 1/n_q^2$. By Assumption 3.3, for sufficiently large $n$ with $\log n \geq 12 \cdot \max(\overline{c}, 1) \cdot (4d + 5)$, there holds

$$\int_{A_j} d\mathrm{P}_X(x') \geq \frac{1}{\tau} \int_{A_j} d\mathrm{Q}_X(x') \geq \frac{1}{\tau}\left(\frac{1}{2^s} - 3 \cdot \frac{1}{6 \cdot 2^s}\right) = \frac{1}{2\tau \cdot 2^s} \tag{35}$$

where we used condition $n_q \geq 2^s \log^2 n$. For the opposite direction, notice that by Assumption 3.3 and 3.2, there holds

$$\int_{A_j} d\mathrm{P}_X(x') \leq \tau \int_{A_j} d\mathrm{Q}_X(x') \leq \frac{\overline{c} \cdot \tau}{2^s}. \tag{36}$$

(35) and (36) together yields the first conclusion. For the second conclusion, again by Lemma C.3, we have

$$\left| \frac{1}{n} \sum_{i=1}^{n} \mathbf{1}\{X_i \in A_j\} - \int_{A_j} d\mathrm{P}_X(x') \right| \leq \sqrt{\frac{\overline{c} \cdot 2^{1-s}(4d+5)\log n}{n}} + \frac{2(4d+5)\log n}{3n} + \frac{4}{n}$$

$$\lesssim \frac{1}{2^s \log n}. \tag{37}$$

for $j \in \mathcal{I}_p$ with probability $\mathrm{P}^n$ at least $1 - 1/n^2$. Consequently, there holds

$$\frac{1}{n} \sum_{i=1}^{n} \mathbf{1}\{X_i \in A_j\} \geq \int_{A_j} d\mathrm{P}_X(x') - \frac{1}{2^s \log n} \geq \frac{1}{2\tau \cdot 2^s} - \frac{1}{4\tau \cdot 2^s} = \frac{1}{4\tau \cdot 2^s}$$

for sufficiently large $n$. Also,

$$\frac{1}{n} \sum_{i=1}^{n} \mathbf{1}\{X_i \in A_j\} \leq \int_{A_j} d\mathrm{P}_X(x') + \frac{1}{2^s \log n} \leq \frac{\overline{c} \cdot \tau}{2^s} + \frac{\overline{c} \cdot \tau}{2^s} = \frac{2\overline{c} \cdot \tau}{2^s}$$

for large $n$. In conclusion, we have

$$\frac{1}{n} \sum_{i=1}^{n} \mathbf{1}\{X_i \in A_j\} \asymp \frac{1}{2^s}. \tag{38}$$

(38) holds when the first conclusion and (37) both hold, which yields a probability $\mathrm{Q}^{n_q} \times \mathrm{P}^n$ at least $1 - 1/n_q^2 - 1/n^2$. $\square$

***Proof of Lemma C.6.*** According to the max-edge partition rule, when the depth of the tree $i$ is a multiple of dimension $d$, each cell of the tree partition is a high-dimensional cube with a side length $2^{-i/d}$. On the other hand, when the depth of the tree $s$ is not a multiple of dimension $d$, we consider the max-edge tree partition with depth $\lfloor i/d \rfloor$ and $\lceil i/d \rceil$, whose corresponding side length of the higher dimensional cube is $2^{-\lfloor i/d \rfloor}$ and $2^{-\lceil i/d \rceil}$. Note that in the splitting procedure of max-edge partition, the side length of each sub-rectangle decreases monotonically with the increase of $s$, so the side length of a random tree partition cell is between $2^{-\lceil i/d \rceil}$ and $2^{-\lfloor i/d \rfloor}$. This implies that

$$\sqrt{d} \cdot 2^{-\lceil i/d \rceil} \leq \mathrm{diam}(A_i^j) \leq \sqrt{d} \cdot 2^{-\lfloor i/d \rfloor}$$

Since $i/d - 1 \leq \lfloor i/d \rfloor \leq \lceil i/d \rceil \leq i/d + 1$, we immediately get $2^{-1}\sqrt{d} \cdot 2^{-i/d} \leq \mathrm{diam}(A_i^j) \leq 2\sqrt{d} \cdot 2^{-i/d}$. $\qquad \square$

***Proof of Lemma C.7.*** For binary tree partition $\pi$, we can grow $\pi$ to a perfect binary tree $\overline{\pi}$ with depth $s$, meaning that all interior nodes have two children and all leaf nodes have the same depth $s$. Then, we can compute the number of leaf nodes on $\overline{\pi}$. On one hand, a depth-$s$ perfect binary tree has $2^s$ leaf nodes [17]. On the other hand, consider the perfect tree grown by $\pi$. Each leaf node $A_j$, when grown to depth $s$, is the root node of a depth-$(s - \mathrm{depth}(A_j))$ perfect binary tree. Thus, the node $A_j$ with induce $2^{s-\mathrm{depth}(A_j)}$ leaf nodes. Immediately, the total number of $\overline{\pi}$ can be computed as $\sum_{j \in \mathcal{I}} 2^{s-\mathrm{depth}(A_j)}$. Matching the results of these two computations yields the desired conclusion. $\qquad \square$

## C.6   Derivation of removing the range parameter

At present, we assume that $\mathrm{P}_X = \mathrm{Q}_X$, but the analogous conclusion holds as long as their density ratio is bounded. Let $X$ be scaled to $\breve{X}$. The excess risk can be decomposed by

$$\mathcal{R}_{L,\mathrm{P}}\left(f_\pi^{\mathrm{DP}}\right) - \mathcal{R}_{L,\mathrm{P}}^* = \int_{[0,1]^d} L(x, y, f_\pi^{\mathrm{DP}}(x))d\mathrm{P}(x,y) - \int_{[0,1]^d} L(x, y, f^*(x))d\mathrm{P}(x,y)$$

$$+ \int_{[0,1]^{dc}} L(x, y, f_\pi^{\mathrm{DP}}(x))d\mathrm{P}(x,y) - \int_{[0,1]^{dc}} L(x, y, f^*(x))d\mathrm{P}(x,y)$$

$$\lesssim \left(\frac{\log n}{n\varepsilon^2}\right)^{\frac{\alpha}{\alpha+d} \wedge \frac{1}{3}} + M^2 \mathbb{P}(\breve{X} \notin \times_{j=1}^d [0,1]).$$

Note that

$$\mathbb{P}(\breve{X} \notin \times_{j=1}^d [0,1]) = \mathbb{P}(X \notin \times_{j=1}^d [\widehat{a^j}, \widehat{b^j}]) \leq \sum_{j=1}^d \mathbb{P}(X^j \notin [\widehat{a^j}, \widehat{b^j}]).$$

Denote the CDF of $X^j$ as $F^j$. Since

$$\mathbb{P}\left(\left|F^j(\widehat{a^j}) - F^j(a^j)\right| > \frac{2\log n_q}{n_q}\right) = \mathrm{P}\left(F^j(\widehat{a^j}) > \frac{2\log n_q}{n_q}\right)$$

$$= \left(1 - \frac{2\log n_q}{n_q}\right)^{n_q} = \frac{1}{n_q^2}$$

and similar result holds for $b^j$, we have $\mathbb{P}(X^j \notin [\widehat{a^j}, \widehat{b^j}]) \lesssim \frac{\log n_q}{n_q}$ with probability $1 - 1/n_q^2$. Thus, there holds $\mathbb{P}(X \notin \times_{j=1}^d [\widehat{a^j}, \widehat{b^j}])) \lesssim \frac{\log n_q}{n_q}$ with probability $1 - d/n_q^2$. Note that if we restrict $n_q \gtrsim (n\varepsilon^2)^{\frac{d \vee 2\alpha}{2\alpha+2d}}$ (instead of $n_q \gtrsim n^{\frac{d}{2\alpha+2d}}$ as in the paper, which is a minor change), there holds $\mathbb{P}(\breve{X} \notin \times_{j=1}^d [0,1]) \lesssim (n\varepsilon^2)^{-\frac{\alpha}{\alpha+d}} \cdot \log n$. Thus, up to a log factor, Theorem 3.4 holds with probability $1 - d/n_q^2$.

# D Experiment details

## D.1 Implementation details

All experiments are conducted on a machine with 72-core Intel Xeon 2.60GHz and 128GB of main memory. The code of LPDT is available on GitHub[2]. Each round of training and testing of LPDT may take less than a second for small data with thousands of samples. For large datasets such as Chicago taxi data, it may take minutes.

- For LPDT-M and LPDT-V, we choose $n_l \in \{2, 5, 10, 20, 40, 60, 80, 100, 120, 140, 160\}$ and $s \in \{1, 2, 3, 4\}$ in Section 4.2. For large data in Section 4.3, we let $s \in \{4, 6, 8, 10\}$. In addition, we add one more parameter adjusting the allocation of the privacy budget on the numerator and denominator. Specifically, let (3) and (4) be

$$\tilde{U}_i^j = \begin{cases} U_i^j - \frac{1}{1+e^{\rho\varepsilon/2}} & \text{with probability } \frac{1}{1+e^{\rho\varepsilon/2}} \\ 1 - U_i^j - \frac{1}{1+e^{\rho\varepsilon/2}} & \text{with probability } \frac{1}{1+e^{\rho\varepsilon/2}}. \end{cases}$$

  as well as

$$\tilde{Y}_i = Y_i + \frac{2M}{(1-\rho)\varepsilon}\xi_i$$

  for $\rho \in [0, 1]$. In this case, the mechanisms are respectively $\rho\varepsilon$-LDP and $(1-\rho)\varepsilon$-LDP, which means the hybrid mechanism is still $\varepsilon$-LDP. We select $\rho \in \{0.3, 0.5, 0.7\}$.

- Decision Tree (DT): For standard non-private decision trees, we use the implementation by Scikit-Learn [47]. We select $max\_depth$ in $\{1, 2, 3, 4\}$ in Section 4.2. For large data in Section 4.3, we let $max\_depth \in \{4, 6, 8, 10\}$.

- Private Histogram (PHIST): We implement the Private Histogram proposed by [9] in Python. PHIST applies the Laplacian mechanism to privatize the estimation of marginal and joint probabilities for a cubic histogram partition with bandwidth $h$. In cells with a marginal probability less than $t$, estimation is truncated to 0. We let $h \in \{1/4, 1/3, 1/2, 1\}$, resulting in the number of grids in each feature to be $\{4, 3, 2, 1\}$. Due to memory limitations, we set $h = 1$ for datasets with $d > 21$. We set the truncation parameter $t \in \{0.01, 0.05\}$.

- Adjusted Private Histogram (APHIST): We implement the Adjusted Private Histogram proposed by [33] in Python. APHIST modifies PHIST by replacing the privatized marginal probability with an average of marginal probability and volume of the cell. In cells with an averaged probability less than $t$, estimation is truncated to 0. We let $h \in \{1/4, 1/3, 1/2, 1\}$, resulting in the number of grids in each feature to be $\{4, 3, 2, 1\}$. Due to memory limitations, we set $h = 1$ for datasets with $d > 21$. We set the truncation parameter $t \in \{0.01, 0.05\}$.

- Deconvolution kernel (DECONV): DECONV imposes noise to $(X_i, Y_i)$ directly to provide privacy. It then treats the regression procedure as a measurement error problem and solves it conventionally using a deconvolution kernel. We implement DECONV proposed by [29] in Python. We set the bandwidth $h \in \{0.02, 0.05, 0.1, 0.2, 0.5, 1, 5\}$.

## D.2 Parameter tuning strategy

To disentangle the parameter tuning problem under local differential, we propose to leave part of the private data as a validation set, whose sample size is $n_v$. For each set of parameters on the parameter grid, the model is partitioned on $X^{pub}$ and trained on $X$. Then the model is passed to each data holder in the validation set. The data holder computes the point-wise error $|\widehat{f}(x) - f(x)|^2$ for each model and returns a privatized value to the curator. By advanced composition [26, 39], for $k$ parameter grids, we can guarantee the $\varepsilon$- privacy of $x$ with a noise $M^2\sqrt{k}\,\xi/\varepsilon$ where $\xi$ is a standard Laplace random variable. Then, the estimated validation mean squared error is $\frac{1}{n_v}\sum_{i=1}^{n_v}\left(\widehat{f}(X_i) - f^*(X_i)\right)^2 + M^2\sqrt{k}\,\xi_i/\varepsilon$. As long as $\sqrt{k}n_v^{-1}\varepsilon^{-1}$ are sufficiently small, the validation procedure can be well conducted. As a result, the capacity of parameter grids is restricted by $o(n_v\varepsilon)$. In many cases, a large amount of data can be acquired if LDP is posed. In such a scenario, we can assure $n_v$ to be large enough to cover the desired parameter grid.

---

[2]https://github.com/Karlmyh/LPDT

## D.3 Details of real data sets

We summarize the details of real data sets in Table 4, with the number of instances and features after pre-processing reported. Each feature is min-max scaled to the range $[0, 1]$ individually. We also present additional information of the data sets including the data source and the pre-processing details.

Table 4: Description of real datasets

| DATASET | $n$ | $d$ | DATASET | $n$ | $d$ |
|---------|-----|-----|---------|-----|-----|
| ABA | 4177 | 8 | CPU | 8192 | 12 |
| AIR | 1503 | 6 | FIS | 908 | 6 |
| ALG | 244 | 12 | HOU | 506 | 13 |
| AQU | 546 | 8 | MUS | 1059 | 68 |
| BUI | 372 | 105 | WHI | 4898 | 12 |
| CBM | 11934 | 16 | RED | 4898 | 12 |
| CCP | 9568 | 4 | CON | 1030 | 8 |

ABA: The *Abalone* dataset originally comes from biological research [44] and now it is accessible on UCI Machine Learning Repository [22]. ABA contains 4177 observations of one target variable and 8 attributes related to the physical measurements of abalone.

AIR: The *Airfoil Self-Noise* dataset on UCI Machine Learning Repository records the result of a series of aerodynamic and acoustic tests of airfoil blade sections conducted in an anechoic wind tunnel [12]. It comprises 1503 instances of 6 attributes including wind tunnel speeds and angles of attack.

ALG: The *Algerian Forest Fires* dataset on UCI Machine Learning Repository contains 244 instances of 11 attributes and 1 output attribute. The task is to predict the condition of forest fires in Algeria [2]. The attribute date is omitted when conducting regression in our experiments.

AQU: The *QSAR aquatic toxicity* dataset was used to develop quantitative regression QSAR models to predict acute aquatic toxicity towards the fish Pimephales promelas (fathead minnow) on a set of 908 chemicals. It contains 546 instances of 8 input attributes and 1 output attribute.

CON: The *Concrete Compressive Strength* dataset on UCI Machine Learning Repository contains 1030 instances of 8 input attributes and 1 output attribute. The task is to predict the concrete compressive strength which is a regression problem.

BUI: The *Residential Building Data Set Data Set* dataset on UCI Machine Learning Repository includes construction cost, sale prices, project variables, and economic variables corresponding to real estate single-family residential apartments in Tehran, Iran. It contains 372 instancesa of 103 input attributes and 2 output attributes.

CBM: The *Condition Based Maintenance of Naval Propulsion Plants* dataset [5] on UCI Machine Learning Repository was generated from a sophisticated simulator of Gas Turbines. It contains 11934 instances of 16 features.

CCP: The *Combined Cycle Power Plant Data Set* dataset [53] on UCI Machine Learning Repository contains 9568 data points. There are 4 features that can be used to predict the net hourly electrical energy output of the power plant.

CPU: The *cpusmall* dataset is from LIBSVM [16]. It contains 8192 instances, each with 12 attributes.

FIS: The *QSAR fish toxicity* dataset on UCI Machine Learning Repository was used to develop quantitative regression QSAR models to predict acute aquatic toxicity towards the fish Pimephales promelas (fathead minnow) on a set of 908 chemicals. It contains 908 instances of 7 features.

RED: This dataset contains the information on red wine of the *Wine Quality* dataset [18] on UCI Machine Learning Repository. There are 11 input variables to predict the output variable wine quality. 4898 instances are collected in the dataset.

WHI: This dataset also originates from the *Wine Quality* dataset [18] on UCI Machine Learning Repository. There are 11 features related to white wine to predict the corresponding wine quality.

HOU: The *Housing-Boston* dataset can be acquired from `LIBSVM` datasets of NTU, which is comprised of 506 observations with 13 features. The dataset is used to predict the price of a house in Boston.

MUS: The *Geographical Original of Music* dataset was built from a personal collection of 1059 tracks covering 33 countries/areas. The program MARSYAS [64] was used to extract audio features from the wave files. We used the default MARSYAS settings in single vector format (68 features) to estimate the performance with basic timbal information covering the entire length of each track.

## D.4 Additional result of the real data experiment

We present the additional result of the real data experiment omitted in the main text due to page limitation.

Table 5: Additional results for average MSE over real data sets for LDP regression methods. The best results are **bolded** and the second best results are underlined. The marked results with significance towards the rest results are marked with ∗. Due to exceeding the memory limit, PHIST and APHIST are corrupted on two data sets which are marked with -.

| | DT | $\varepsilon = 0.5$ | | | | | $\varepsilon = 1$ | | | | |
|---|---|---|---|---|---|---|---|---|---|---|---|
| | | LPDT-M | LPDT-V | APHIST | PHIST | DECONV | LPDT-M | LPDT-V | APHIST | PHIST | DECONV |
| ABA | 5.67e+0 | 1.10e+1 | **1.07e+1** | 1.12e+1 | 1.16e+1 | 1.05e+7 | 1.06e+1 | **1.01e+1*** | 1.06e+1 | 1.07e+1 | 3.06e+6 |
| AIR | 2.26e+1 | 3.79e+2 | **1.29e+2*** | 2.10e+2 * | 3.10e+2 | 1.19e+6 | 1.67e+2 | 1.17e+2* | **9.25e+1*** | 1.26e+2 | 5.64e+7 |
| ALG | 2.12e-2 | **2.59e-1*** | **2.59e-1*** | 2.74e-1 | 2.83e-1 | 8.81e+2 | **2.40e-1*** | 2.40e-1* | 2.56e-1 | 2.59e-1 | 1.73e+2 |
| AQU | 1.92e+0 | 1.35e+1 | **3.01e+0*** | 3.68e+0* | 4.06e+0 | 6.02e+4 | 3.06e+0 | 3.63e+0 | **2.99e+0*** | 3.11e+0 | 9.55e+3 |
| BUI | 1.75e+5 | 3.54e+6* | **1.70e+6*** | - | - | 1.20e+9 | 2.46e+6* | **1.54e+6*** | - | - | 3.30e+9 |
| CBM | 4.08e-27 | 6.96e+0 | **6.64e+0*** | 6.96e+0* | 6.99e+0 | 9.69e+6 | 2.46e+0 * | **1.90e+0*** | 6.91e+0 | 6.92e+0 | 1.58e+6 |
| CCP | 2.19e+1 | **4.52e+2*** | 4.89e+2* | 6.79e+2 | 9.31e+2 | 1.23e+10 | 4.72e+2 | 4.72e+2 | **4.15e+2*** | 4.82e+2 | 1.43e+7 |
| CON | 9.38e+1 | 4.56e+2 | 4.56e+2 | **3.24e+2*** | 3.38e+2* | 3.33e+8 | 3.08e+2 | 3.42e+2 | **2.99e+2** | 3.03e+2 | 3.18e+7 |
| CPU | 2.15e+1 | **3.56e+2** | 3.56e+2 | 3.56e+2 | 3.68e+2 | 6.61e+7 | 3.41e+2 | **9.37e+1*** | 3.41e+2 | 3.43e+2 | 9.37e+7 |
| FIS | 1.07e+0 | 3.66e+0 | 3.66e+0 | **2.62e+0*** | 2.81e+0* | 1.68e+5 | 2.48e+0 | 2.48e+0 | **2.24e+0*** | 2.30e+0 | 1.57e+4 |
| HOU | 2.11e+1 | 1.05e+2* | **1.04e+2** | 1.08e+2 | 1.17e+2 | 2.34e+5 | **8.56e+1*** | 8.56e+1* | 9.00e+1* | 9.32e+1 | 1.55e+7 |
| MUS | 3.00e+2 | **6.21e+2*** | 6.21e+2* | - | - | 2.18e+5 | **4.26e+2*** | 4.26e+2* | - | - | 1.29e+6* |
| RED | 4.76e-1 | **8.78e-1*** | 8.94e-1* | 1.09e+0 | 1.30e+0 | 2.74e+6 | 7.67e-1* | **7.04e-1*** | 8.01e-1 | 8.69e-1 | 1.15e+5 |
| WHI | 5.77e-1 | 8.77e-1* | **8.69e-1*** | 9.45e-1 | 1.03e+0 | 1.28e+9 | 8.68e-1 | 8.48e-1 | **8.32e-1*** | 8.56e-1 | 1.58e+6 |

| | DT | $\varepsilon = 4$ | | | | | $\varepsilon = 8$ | | | | |
|---|---|---|---|---|---|---|---|---|---|---|---|
| | | LPDT-M | LPDT-V | APHIST | PHIST | DECONV | LPDT-M | LPDT-V | APHIST | PHIST | DECONV |
| ABA | 5.67e+0 | **9.46e+0*** | 9.54e+0* | 2.00e+1 | 1.05e+1 | 1.14e+1 | 7.96e+0* | **6.94e+0*** | 2.07e+1 | 1.05e+1 | 1.05e+1 |
| AIR | 2.26e+1 | 4.67e+1* | **4.15e+1*** | 1.52e+3 | 5.28e+1 | 4.85e+1 | 4.45e+1* | **3.47e+1*** | 1.64e+3 | 4.87e+1 | 4.71e+1 |
| ALG | 2.12e-2 | **2.40e-1*** | 2.42e-1* | 2.59e-1 | 2.48e-1 | 2.03e+3 | **2.43e-1** | 2.46e-1 | 2.64e-1 | 2.46e-1 | 2.54e-1 |
| AQU | 1.92e+0 | 2.81e+0 | **2.77e+0*** | 4.54e+0 | 2.85e+0 | 5.23e+0 | 2.71e+0* | **2.63e+0*** | 4.86e+0 | 2.83e+0 | 2.85e+0 |
| BUI | 1.75e+5 | 1.49e+6* | **1.35e+6*** | - | - | 3.90e+7 | 1.40e+6* | **1.28e+6*** | - | - | 1.38e+7 |
| CBM | 4.08e-27 | 9.34e-1* | **5.10e-1*** | 7.45e+0 | 6.48e+0 | 1.41e+5 | 7.08e-1* | **3.77e-2*** | 4.49e+0 | 2.00e+0 | 6.44e+0 |
| CCP | 2.19e+1 | 9.08e+1* | **6.78e+1*** | 2.20e+4 | 3.34e+2 | 2.86e+2 | 8.22e+1* | **4.76e+1*** | 2.26e+4 | 3.26e+2 | 2.10e+2 |
| CON | 9.38e+1 | 2.70e+2* | **2.54e+2*** | 4.06e+2 | 2.97e+2 | 3.93e+2 | 2.34e+2* | **2.04e+2*** | 4.21e+2 | 2.96e+2 | 2.91e+2 |
| CPU | 2.15e+1 | 3.05e+2* | **8.36e+1*** | 9.65e+2 | 3.40e+2 | 3.93e+5 | 3.01e+2* | **4.85e+1*** | 1.02e+3 | 3.40e+2 | 3.42e+2 |
| FIS | 1.07e+0 | **1.90*** | 2.07e+0 | 3.47e+0 | 2.17e+0 | 2.40e+0 | **1.55e+0*** | 1.65e+0* | 3.67e+0 | 2.15e+0 | 2.17e+0 |
| HOU | 2.11e+1 | 7.80e+1* | **7.37e+1*** | 1.18e+2 | 8.27e+1 | 1.31e+3 | 7.32e+1* | **7.02e+1*** | 1.26e+2 | 8.19e+1 | 1.27e+2 |
| MUS | 3.00e+2 | **3.33e+2*** | 3.46e+2* | - | - | 8.03e+4 | **3.21e+2*** | 3.26e+2* | - | - | 2.25e+4 |
| RED | 4.76e-1 | 7.13e-1 | **6.92e-1** | 3.63e+0 | 7.19e-1 | 4.03e+2 | 6.14e-1* | **5.79e-1*** | 3.88e+0 | 7.09e-1 | 6.94e-1 |
| WHI | 5.77e-1 | **7.39e-1*** | 7.50e-1* | 4.34e+0 | 8.05e-1 | 1.88e+4 | 6.93e-1* | **6.40e-1*** | 4.50e+0 | 8.02e-1 | 8.21e-1 |

## D.5 Details of Chicago taxi dataset

The dataset used in this study was obtained from the Differential Privacy Temporal Map Challenge (DeID2), which aims to develop algorithms that preserve data utility while guaranteeing individual privacy protection. The dataset contains quantitative and categorical information about taxi trips in Chicago, including time, distance, location, payment, and service provider. These features include the identification number of taxis ($taxi\_id$), time of each trip ($seconds$), the distance of each trip ($miles$), index of the zone where the trip starts ($pca$), index of the zone where the trip ends ($dca$), service provider ($company$), the method used to pay for the trip ($payment\_type$), amount of tips ($tips$), and amount of fares ($fare$).

To preprocess the data, we selected locations in the central region of the map data to ensure sufficient passenger pick-up and drop-off orders, and we chose the top 10 taxi companies with the highest number of orders. We then divided the dataset based on the payment feature to distinguish between

Table 6: Average time over real data sets for LDP regression methods. Due to exceeding the memory limit, PHIST and APHIST are corrupted on two data sets which are marked with -.

|  | DT | LPDT-M | LPDT-V | APHIST | PHIST | DECONV |
|---|---|---|---|---|---|---|
| ABA | 5.34e-3 | 1.78e-2 | 3.23e-2 | 2.70e-2 | 2.71e-2 | 1.47e+0 |
| AIR | 2.63e-3 | 1.13e-2 | 1.76e-2 | 9.68e-3 | 9.58e-3 | 1.29e-1 |
| ALG | 1.79e-3 | 2.02e-3 | 2.81e-3 | 6.10e-3 | 5.55e-3 | 1.17e-2 |
| AQU | 2.21e-3 | 3.38e-3 | 5.21e-3 | 7.17e-3 | 5.62e-3 | 3.12e-2 |
| BUI | 3.87e-3 | 4.09e-3 | 2.61e-2 | - | - | 1.35e-1 |
| CBM | 2.58e-2 | 8.98e-2 | 1.23e-1 | 1.04e-1 | 1.04e-1 | 2.76e+1 |
| CCP | 8.11e-3 | 6.28e-2 | 6.98e-2 | 4.65e-2 | 4.70e-2 | 4.42e+0 |
| CON | 3.28e-3 | 6.38e-3 | 1.01e-2 | 7.45e-3 | 7.14e-3 | 9.06e-2 |
| CPU | 1.52e-2 | 3.99e-2 | 6.40e-2 | 5.86e-2 | 5.80e-2 | 9.75e+0 |
| FIS | 3.00e-3 | 4.53e-3 | 7.63e-3 | 6.64e-3 | 6.01e-3 | 5.53e-2 |
| HOU | 2.52e-3 | 3.37e-3 | 5.30e-3 | 5.21e-3 | 5.54e-3 | 5.03e-2 |
| MUS | 6.24e-3 | 6.64e-3 | 4.94e-2 | - | - | 6.49e-1 |
| RED | 3.49e-3 | 9.01e-3 | 1.41e-2 | 1.17e-2 | 1.18e-2 | 2.65e-1 |
| WHI | 7.16e-3 | 1.68e-2 | 4.26e-2 | 3.29e-2 | 3.29e-2 | 2.70e+0 |

Table 7: Additional results for average MSE over real data sets for decision trees. The best results are **bolded**.

|  | DT | DT-pub | PDT-M | PDT-V |
|---|---|---|---|---|
| ABA | 5.67e+0 | 7.93e+0 | 8.38e+0 | 7.34e+0 |
| AIR | 2.26e+1 | 4.28e+1 | 4.49e+1 | 3.60e+1 |
| ALG | 2.12e-2 | 2.59e-1 | 2.44e-1 | 2.46e-1 |
| AQU | 1.92e+0 | 2.42e+0 | 2.73e+0 | 2.67e+0 |
| BUI | 1.75e+5 | 1.39e+6 | 1.44e+6 | **1.31e+6** |
| CBM | 4.11e-27 | 1.57e+0 | 7.62e-1 | **1.23e-1** |
| CCP | 2.19e+1 | 7.21e+1 | 8.42e+1 | **5.18e+1** |
| CON | 9.38e+1 | 2.21e+2 | 2.44e+2 | **2.13e+2** |
| CPU | 2.10e+1 | 2.92e+2 | 3.02e+2 | 6.15e+1 |
| FIS | 1.07e+0 | 1.48e+0 | 1.65e+0 | 1.76e+0 |
| HOU | 2.11e+1 | **5.58e+1** | 7.43e+1 | 7.10e+1 |
| MUS | 3.00e+2 | 3.38e+2 | 3.27e+2 | **3.27e+2** |
| RED | 4.76e-1 | **5.69e-1** | 6.75e-1 | 6.12e-1 |
| WHI | 5.76e-1 | 7.10e-1 | 7.03e-1 | **6.61e-1** |

private and public data, which is in accordance with the fact that we only had access to specific data holders. Additionally, we converted all categorical features, including $pca$, $dca$, and $company$, into one-hot values, resulting in a 102-dimensional feature vector. Ultimately, we made predictions for the fares of the entire trip.

Figure 6 shows a boxplot of the prediction feature in the Chicago taxi dataset. As we can see from the plot, the public data and private data differ in distribution, which we referred to as a distribution shift in the main text.

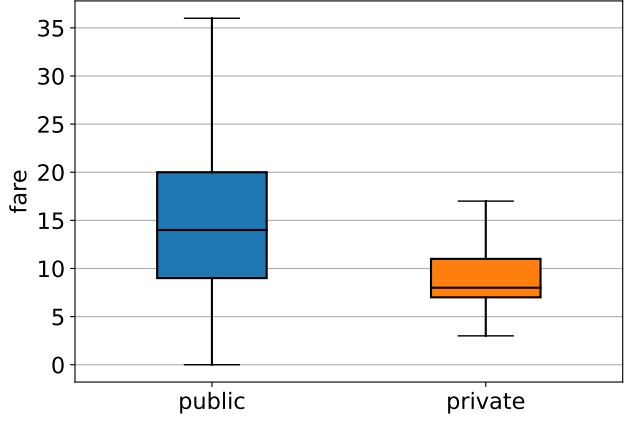

Figure 6: Boxplot of fare in Chicago taxi dataset

