# OpenReview forum: "Decision Tree for Locally Private Estimation with Public Data"
_NeurIPS.cc/2023/Conference — NeurIPS 2023 poster_

### Official Review · Reviewer_HsA8 · 2023-06-26

**Soundness:** 3 good
**Presentation:** 3 good
**Contribution:** 3 good
**Rating:** 6
**Confidence:** 3

**Summary:**

In this paper, the authors propose a scheme for training decision trees on a combination of public and private data. By using local differential privacy the algorithm guarantees that adversaries learn little about the private data from the outcomes of the algorithm. Under assumptions and a specific splitting rule for the algorithm's public part, the authors prove properties about the convergence and generalization of the algorithm. The algorithm is empirically evaluated on 15 real and 1 synthetic dataset and compared against private histograms and deconvolution.

**Strengths:**

- The paper is written in clear English.
- Substantial theoretical analysis.
- Empirical evaluation on many (15) datasets shows that this method outperforms private histograms and deconvolution.
- The effect of different hyperparameter settings was clearly explored.

**Weaknesses:**

Major:
- Assumption 3.2 is key to deriving error bounds but I am unsure if this holds. Decision trees are discontinuous models that are often used to model discontinuous data.
- Sections 2 and 3 are very dense in mathematical notation and use complicated phrasing to say simple things. E.g. section 2.2 spends a full page that is hard to read on explaining that: an estimated regression tree predicts the mean of samples in a leaf and that the private tree predicts it based on private estimates of reaching the leaf (U) and the sample value (Y), privatized using randomized response and Laplace mechanism.
- Only relatively high values of $\epsilon$ were explored. While it is debatable what is a good value, to the best of my knowledge $\epsilon < 1$ is generally considered 'good privacy'.

Minor:
- Although I understand the 'max-edge partition with variance reduction' was introduced to prove theoretical properties it suffers from data that is non-uniformly distributed or contains useless features.
    - Useless features: the algorithm will never choose the same feature to split on twice unless depth > d (as a result of splitting on the largest distance). This means that if there are few informative features these will be split on only once.
    - Non-uniform data: the splits are determined on the midpoint between minimum and maximum values of a feature instead of looking at how data is distributed in the feature. This means data with e.g. long tails will cause a split to create one leaf with almost all data and one leaf with almost none.
- The algorithm and analyses are based on continuous features but the datasets used for evaluation also contain categorical features. E.g. 'sex' in Abalone.
- Limitations and broader impact have been moved to the appendix while these should be part of the main text.
- The visualizations only show mean estimates, no standard deviations / standard errors.
- It would be nice to compare to (global) differentially private decision trees to get an idea of the cost of decentralization in this task.

**Questions:**

1. Does assumption 3.2 hold when considering decision trees that make discontinuous partitions?
2. Does the proposed algorithm still outperform existing work when considering better privacy protection with $\epsilon < 1$?
3. Can the theoretical guarantees be applied to categorical data (where distances such as 'diam' do not make sense)?

**Limitations:**

The limitations are moved to the appendix with the only limitation discussed being assumption 3.3.

Depending on the answer to question 1 I believe assumption 3.2 needs to be added to the limitations and a discussion on the choices of $\epsilon$ should be added. The limitations should be given in the main text.

---

> ### Author Rebuttal · Authors · 2023-08-10
>
>
> Q1:
> Assumption 3.2 is key to deriving error bounds but I am unsure if this holds. Does assumption 3.2 hold when considering decision trees that make discontinuous partitions?
>
> A1:
> The reviewer is correct.
> We apologize for the misleading statement of Assumption 3.2.
> Assumption 3.2 is posed on the true regression function $f^*(x) = \mathbb{E} [Y|X = x]$ instead of any regression function $f$.
> Thus, the assumption is irrelevant to the fitted decision tree and holds for any estimator.
>
>
> Q2:
> Sections 2 and 3 are very dense in mathematical notation and use complicated phrasing.
>
> A2:
> We apologize for the dense mathematical notations.
> These definitions and writing flows (defining the population estimator and its empirical estimate) are broadly seen in nonparametric statistics.
> We will revise to avoid complicated notations.
>
> Q3:
> Only relatively high values of $\varepsilon$ were explored.
>
> A3: Please see R1 in the global response.
>
>
> Q4:
> Although I understand the 'max-edge partition with variance reduction' was introduced to prove theoretical properties it suffers from data that is non-uniformly distributed or contains useless features.
>
>
> A4:
> We thank the reviewer for the valuable and surprisingly insightful comment.
> In practice, we find that the max-edge rule actually works well given certain preprocess procedures.
> i) uniformizing (for each feature, map the value of each sample to the empirical CDF).
> After uniformizing, the data is approximately uniformly distributed along each feature and the issue caused by non-uniformly distribution is legitimated.
> ii) PCA/feature selection. The useless features can be eliminated by the dimension reduction technique.
> PCA is recommended. Feature selection may retain some binary features, when $p> 2d$, the binary features will be split twice, which is meaningless.
>
>
>
> Q5: The algorithm and analyses are based on continuous features.
> Can the theoretical guarantees be applied to categorical data (where distances such as 'diam' do not make sense)?
>
> A5:
> In the context of decision tree studies, this is a common gap between theory and experiment.
> The theoretical framework applies only to continuous features.
> Meanwhile, decision trees are well suited for dealing with categorical features in practice.
>
> Q6:
> Limitations and broader impact have been moved to the appendix while these should be part of the main text.
> The limitations are moved to the appendix with the only limitation discussed being assumption 3.3.
> Depending on the answer to question 1 I believe assumption 3.2 needs to be added to the limitations and a discussion on the choices
>  should be added. The limitations should be given in the main text.
>
>
> A6:
> We apologize for doing so due to page limitations.
> We will add discussions about other assumptions as well.
>
>
> Q7: The visualizations only show mean estimates, no standard deviations / standard errors.
>
>
> A7: We will add the error bar to the figures as shown in the PDF.
> But we do not have time to create well-presented figures for all experiments during rebuttal time.
>
> Q8: It would be nice to compare to (global) differentially private decision trees to get an idea of the cost of decentralization in this task.
>
> A8:
> To the best of our knowledge, there is no existing private decision tree algorithm for regression.
> But generally, central DP models perform much better than LDP models under the same privacy budget.
>
>
>
>
>
>
> Q9: Does the proposed algorithm still outperform existing work when considering better privacy protection with?
>
> A9: Yes, please see R1 and R3 in the global response.

---

> ### Author Response · Authors · 2023-08-20
>
> We are wondering if the reviewer receives a satisfying answer from the author's rebuttal and wants to reconsider the rating.
> We are happy to answer any other questions but please leave enough time for us to answer them, as the discussion period deadline is approaching.

---

> > ### Comment · Reviewer_HsA8 · 2023-08-20
> >
> > My apologies for the late response and thank you for answering my questions. I will increase my rating of the submission given that the clarifications make it to the paper. I generally agree with the rebuttal and I only have two minor comments:
> >
> > Regarding A4, using techniques such as PCA in combination with decision trees limit the interpretability of the models as the features are no longer easy to understand. Therefore I would not rely on too many pre-processing techniques to improve performance. However, this was a minor point and the results show that the method works sufficiently well regardless.
> >
> > I would not rely on the epsilon values used by companies as a guideline for good privacy as values much larger than 2 have also been used in practice. In my opinion, the paper should have results with low epsilon values as well (such as in the rebuttal PDF), even if it shows that all methods fail to perform well.

---

> > > ### Author Response · Authors · 2023-08-21
> > >
> > > Thank you for the valuable comments. We will include them as part of our discussions. Indeed, the experiments for small $\varepsilon$ are important, and the results will be incorporated into the main context.

---

### Official Review · Reviewer_xd2A · 2023-07-03

**Soundness:** 3 good
**Presentation:** 3 good
**Contribution:** 2 fair
**Rating:** 5
**Confidence:** 3

**Summary:**

The paper studies nonparametric regression under local differential privacy (LDP) constraints with the aid of public data. The paper proposes an algorithm, locally differentially private decision tree (LPDT), which uses public data to construct the splitting criteria for the decision tree and private data to compute the regressed values for each leaf node. The paper shows that under certain assumptions, with a small amount of public data, the algorithm can achieve a near-optimal convergence rate for decision tree regression under LDP constraints. The paper also demonstrates the effectiveness of the algorithm through experiments.

**Strengths:**

1. The paper studies an interesting problem, which has not been studied before to the best of my knowledge. The paper also proposes an algorithm that achieves an optimal convergence rate under reasonable assumptions.

2. The paper shows that without public data, decision tree learning will have a nontrivial risk with any amount of private data. This suggests the importance of public data for decision tree regression under LDP constraints.

**Weaknesses:**

1. The importance of public data has been established for different learning tasks recently, especially under the central notion of differential privacy. Extending a similar observation to decision tree learning under local DP constraints is interesting, but less surprising.

2. The similarity assumption between P and Q, Assumption 3.3, looks weird. The assumption doesn't pose any requirement on how y is correlated with x. Imagine the case when P and Q have the same marginal on the feature space while the correlation between X and Y are completely different under P and Q. It would be hard to learn useful information about how x can be used to predict y from public data, and hence the structure of a good decision tree. I am not sure whether I have missed other important assumptions.

3. Another baseline to compare (theoretically, and empirically) would be to only use public data to train the decision tree. How would the proposed method compare against this baseline? E.g., in the experiment in Section 4.2.

**Questions:**

See the questions above.

Additional questions:
1. In Theorem 3.4, how does the error depend on tau? Could you comment on the dependence on tau as well?
2. Figure 4(b), when delta = 0, and eps is large, does the proposed method recover the non-private training rate?
3. For LDP estimation, the dependence on eps can be improved to inverse exponential in eps when eps is large, e.g. in Feldman et al 2022. Can the rate in Theorem 3.4 improve to a similar dependence on eps as well? It seems the considered eps's in the experimental section are mostly large.

[Feldman et al 2022] Vitaly Feldman, Jelani Nelson, Huy Nguyen, Kunal Talwar. Private Frequency Estimation via Projective Geometry

**Limitations:**

Yes.

---

> ### Author Rebuttal · Authors · 2023-08-10
>
>
> Q1: The importance of public data has been established  ... but less surprising.
>
> A1:
> We agree with the reviewer that the idea of using public data is less surprising.
> However, we would like to emphasize our motivation for utilizing public data.
> Most public data usage, such as gradient preconditioning or public pretraining, uses public data to accelerate convergence.
> Our public data is used for adaptive partitioning, which is impossible under LDP constraints without public data.
> As a result, we claim our contribution as using public data to conduct procedures that would be infeasible without access to the raw private data.
>
>
> Q2: The similarity assumption between P and Q, Assumption 3.3, looks weird ... whether I have missed other important assumptions.
>
> A2:
> The reviewer is correct. Please see R2 in the global response.
> Without posing strong assumptions, utilizing information in $Y|X$ is hard even in the non-private case [2,3] and can be regarded as future research directions.
>
> Q3: Another baseline to compare (theoretically, and empirically) would be to only use public data ... in the experiment in Section 4.2.
>
> A3:
> In the setting of Section 4.2, the public and private distributions are identical.
> In this case, theoretically, using $n_q \gtrsim (n\varepsilon^2)^{\frac{2\alpha + d}{2\alpha + 2d}}$ public samples can be more effective that LPDT.
> Empirically, we conduct experiments with respect to such baseline in R1 of the global response.
> The result shows that, when the privacy level is low and the public data is insufficient, LPDT is still the best choice.
>
>
> Moreover, we would like to argue that, once under different public and private distributions, using public data will fall both theoretically and empirically.
>
>
>
>
> Q4: In Theorem 3.4, how does the error depend on tau? Could you comment on the dependence on tau as well?
>
> A4: When the parameter $\tau$ is not treated as a constant, the upper bound becomes $(\frac{\tau^2\log n}{n \varepsilon^2})^{\frac{\alpha}{\alpha+d} \wedge \frac{1}{3}}$.
> The smaller $\tau$ is, the closer $\mathrm{P}_X$ and $\mathrm{Q}_X$ are, the tighter the upper bound gets.
>
> Q5:
> Figure 4(b), when delta = 0, and eps is large, does the proposed method recover the non-private training rate?
>
> A5:
> The MSE for $\delta = 0$ and $\varepsilon = 1000$ is 1.11.
> There is a slight performance gap caused by creating the partition from one dataset and fit on the other.
> But generally, the MSE matches the performance of training a decision tree directly on private data.
>
>
> Q6:
> For LDP estimation, the dependence on eps ... the considered eps's in the experimental section are mostly large.
>
> A6:
> Technically speaking, under the proposed mechanisms, we do not think the improvement of the upper bound is possible.
> The Laplace mechanism necessitates the inclusion of a $\varepsilon^2$ term and cannot be changed to $(e^\varepsilon - 1)^2$. Conversely, the lower bound is drawn using techniques in [4] and must include a factor of $(e^\varepsilon - 1)^2$. However, we propose the following improvements to fix the gap between theory and empirical findings.
>
> 1. We present the experiment results for small $\varepsilon$ as in R3.
>
> 2. An upper bound that matches the lower bound can be derived by introducing a novel mechanism, as described below. Upon partitioning $\mathcal{X}$, a corresponding partition is created for $\mathcal{Y}$ (done uniformly or using information from public data), denoted as ${B_1, B_2, \ldots, B_k, \ldots}$. Subsequently, each data holder generates one-hot encodings $V_i^{j,k} = \mathbf{1}{X_i \in A_j, Y_i \in B_k}$ and $U_i^j$ as in the paper. These quantities are privatized using random responses and can be utilized to estimate equation (1) in the paper. The use of random response can improve the convergence rate to $(n(e^{\varepsilon} - 1)^2)^{-\frac{\alpha}{\alpha + d}}$. Nevertheless, the new mechanism might be less effective in practical applications. Due to the substantial number of grids, only a limited number of samples exist within each $A_j \times B_k$, resulting in high instability during estimation within each grid.
>
>
> [2] T Tony Cai and Hongming Pu. Transfer learning for nonparametric regression: Non-asymptotic minimax analysis and adaptive procedure. arXiv preprint arXiv:0000.0000, 2022.
>
> [3] T Tony Cai and Hongji Wei. Transfer learning for nonparametric classification: Minimax rate and adaptive classifier. 2021.

---

> > ### Comment · Reviewer_xd2A · 2023-08-14
> >
> > Thanks the authors for their detailed response. I have one more question:
> >
> > Regarding Q3 & A3, I couldn't find the stated results comparing against the baseline with only public data. Could the authors give a clearer pointer?

---

> > > ### Author Response · Authors · 2023-08-15
> > >
> > > We apologize for the mistake. The results were supposed to be included in Table 1, and we now present the results here.
> > > We train a decision tree on public data (whose sample size is 10\% of the training set, i.e. $n_q = 0.1 (n+n_q)$.
> > > It is compared with the decision tree trained on private data and LPDT with $\varepsilon = 6$.
> > > The best results are bolded.
> > > The results show that when $n_q\ll n$ and $\varepsilon$ is large, training a decision tree on public data is worse than using LPDT on most of the datasets.
> > >
> > > |  | **DT**   | **DT-pub** | **PDT-M** | **PDT-V** |
> > > |:-------------:|:--------:|:------------:|:---------:|:----------:|
> > > | **ABA**       | 5.67e+0  | 7.93e+0      | 8.38e+0   | **7.34e+0**    |
> > > | **AIR**       | 2.26e+1  | 4.28e+1      | 4.49e+1   | **3.60e+1**    |
> > > | **ALG**       | 2.12e-2  | 2.59e-1      | **2.44e-1**   | 2.46e-1    |
> > > | **AQU**       | 1.92e+0  | **2.42e+0**      | 2.73e+0   | 2.67e+0    |
> > > | **BUI**       | 1.75e+5  | 1.39e+6      | 1.44e+6   | **1.31e+6**    |
> > > | **CBM**       | 4.11e-27 | 1.57e+0      | 7.62e-1   | **1.23e-1**    |
> > > | **CCP**       | 2.19e+1  | 7.21e+1      | 8.42e+1   | **5.18e+1**   |
> > > | **CON**       | 9.38e+1  | 2.21e+2      | 2.44e+2   | **2.13e+2**    |
> > > | **CPU**       | 2.10e+1  | 2.92e+2      | 3.02e+2   | **6.15e+1**    |
> > > | **FIS**       | 1.07e+0  | **1.48e+0**      | 1.65e+0   | 1.76e+0    |
> > > | **HOU**       | 2.11e+1  | **5.58e+1**      | 7.43e+1   | 7.10e+1    |
> > > | **MUS**       | 3.00e+2  | 3.38e+2      | 3.27e+2   | **3.27e+2**    |
> > > | **RED**       | 4.76e-1  | **5.69e-1**      | 6.75e-1   | 6.12e-1    |
> > > | **WHI**       | 5.76e-1  | 7.10e-1      | 7.03e-1   | **6.61e-1**    |

---

> > > > ### Comment · Reviewer_xd2A · 2023-08-16
> > > >
> > > > Thanks for the response. I still think the paper is borderline. Hence I tend to keep my score.

---

### Official Review · Reviewer_QZtg · 2023-07-05

**Soundness:** 3 good
**Presentation:** 3 good
**Contribution:** 2 fair
**Rating:** 6
**Confidence:** 3

**Summary:**

This work studies non-parametric estimation under local differential privacy with public data. The authors propose a locally private decision tree algorithm and show that it is min-max optimal under some regimes. The authors also test the algorithm on real and synthetic datasets.

**Strengths:**

1. The idea to leverage public data in LDP estimation is practically relevant.
2. The experiment results are extensive. It includes both synthetic and real-world data and discussion on the effect of various parameters. Empirically the algorithm outperforms existing algorithms in most experiment settings.
3. The authors provide a thorough theoretical analysis of the proposed algorithm in terms of sample complexity and computation cost.  The proposed algorithm has improved time complexity over prior works.

**Weaknesses:**

1. The proposed algorithm does not seem to improve too much in terms of sample complexity with public data. We normally hope that using public data should improve the performance somehow (e.g. in reference [8], with public data the prior bound on the mean can be removed), but it seems that the algorithm merely attains the optimal bound without public data only when $\varepsilon\lesssim 1$. Existing algorithms that do not use public data can also achieve the same sample complexity.
2. Given the first point, the fact that the proposed algorithm does not work without public data appears to be a significant disadvantage. While LPDT has advantages in time complexity and empirically outperforms existing methods, it comes at the expense of additional resource of public data. It is thus in some sense unfair to compare with existing methods that do not use public data.

#### Some minor problems

1. The lower bound only matches the upper bound in Theorem 3.4 when $\varepsilon\lesssim 1$ (since $e^{\varepsilon}-1\simeq \varepsilon$ only with small $\varepsilon$), but all experiment results are for large $\varepsilon\ge 2$.
2. The notation $p$ is used first for the density function and then again used for decision tree depth. Please consider changing one of them to avoid confusion.

#### Update
- Increased my score to 5 after the authors added new results for small $\varepsilon$ and introduced a new algorithm that matches the lower bound for all ranges of $\varepsilon$.
- Increased to 6 after the authors resolved my question for removing range parameters.

**Questions:**

Can existing algorithms be easily modified to take advantage of public data? I feel it is best to fully investigate the possibility of leveraging existing algorithms before inventing something entirely new. Extension based on prior algorithms also serve as a necessary baseline to compare against.


**Limitations:**

The authors adequately addressed the limitations and potential negative societal impact.

---

> ### Author Rebuttal · Authors · 2023-08-10
>
> Q1: The proposed algorithm does not seem to improve too much ... the same sample complexity.
>
> A1: Please see the R2 of the global response for the theoretical improvements.
>
>
> Q2: Given the first point, the fact that the proposed algorithm does not work without public data appears to be a significant disadvantage. While LPDT has advantages in time complexity and empirically outperforms existing methods, it comes at the expense of additional resources of public data. It is thus in some sense unfair to compare with existing methods that do not use public data.
>
> A2: We agree with the reviewer that the additional public data can be an unfair comparison.
> Given both private data and public data, one has three choices: leveraging only private data, only public data, or both.
> Previous methods focus on the first choice, while the second choice is prohibited since we allow $\mathrm{P}$ and $\mathrm{Q}$ to be completely different.
> When $E_P[Y|X] = E_Q[Y|X] + 1$, this could lead to a trivial risk.
> Our method explores the third choice for the first time and works well as long as a small amount of public data exists.
> For better illustration, we conduct an additional set of experiments (Experiment 4 in the global response) to show that, to attain similar performance, the sample size $n$ and $n_q$ required by LPDT can be much smaller than previous methods.
> The observation suggests that, when data-collecting resources are limited, organizations can put up rewards to get a small amount of public data rather than collecting a large amount of private data.
>
>
> Q3: The lower bound only matches the upper bound in Theorem 3.4 ...
>
> A3: We propose the following improvements to fix the gap between theory and empirical findings.
>
> 1. We present the experiment results for small $\varepsilon$ as in R3.
>
> 2. An upper bound that matches the lower bound can be derived by introducing a novel mechanism, as described below. Upon partitioning $\mathcal{X}$, a corresponding partition is created for $\mathcal{Y}$ (done uniformly or using information from public data), denoted as ${B_1, B_2, \ldots, B_k, \ldots}$. Subsequently, each data holder generates one-hot encodings $V_i^{j,k} = \mathbf{1}{X_i \in A_j, Y_i \in B_k}$ and $U_i^j$ as in the paper. These quantities are privatized using random responses and can be utilized to estimate equation (1) in the paper. The use of random response can improve the convergence rate to $(n(e^{\varepsilon} - 1)^2)^{-\frac{\alpha}{\alpha + d}}$. Nevertheless, the new mechanism might be less effective in practical applications. Due to the substantial number of grids, only a limited number of samples exist within each $A_j \times B_k$, resulting in high instability during estimation within each grid.
>
>
> Q4:  The notation  $p$ is used first for the density function ... to avoid confusion.
>
> A4: We apologize for the misuse of notations. We will use $s$ to denote the depth.
>
> Q5:
> Can existing algorithms be easily modified to take advantage of public data? I feel it is best to fully investigate the possibility of leveraging existing algorithms before inventing something entirely new. Extension based on prior algorithms also serves as a necessary baseline to compare against.
>
> A5:
> The reviewer brings up an interesting question.
> One of the potential strategies is to form a private estimator from $\mathrm{P}$ data and a normal estimator from $Q$ data. Then one can take the weighted average of the estimators, following [3], or adaptively choose one of them, following [2].
> Yet, both approaches require additional assumptions on the regression functions $P_{Y|X}$ and $Q_{Y|X}$.
> We hope the reviewer can provide some clues about how the existing methods can be modified.
>
>
> [2] T Tony Cai and Hongming Pu. Transfer learning for nonparametric regression: Non-asymptotic minimax analysis and adaptive procedure. arXiv preprint arXiv:0000.0000, 2022.
>
> [3] T Tony Cai and Hongji Wei. Transfer learning for nonparametric classification: Minimax rate and adaptive classifier. 2021.

---

> > ### Comment · Reviewer_QZtg · 2023-08-15
> > **Thank you for your response**
> >
> > Thank you for your response. You have proposed a new algorithm that matches the lower bound and added experiments and discussions for small $\varepsilon$ which addressed some of my concerns. I am willing to increase my score to 5 for now.
> >
> > I still have questions about the improvement over no public data. You mentioned that "with high probability" your algorithm can remove the requirement on the range parameter, similar to Bie et.al. 2022 (reference [8] in the paper). However, since the argument is not elaborated in the paper, could you please give a more detailed and precise argument about how the range requirement can be removed and what exactly are the high probability guarantees? I am willing to increase my score to 6 if you could further provide a more concrete argument.

---

> > > ### Author Response · Authors · 2023-08-15
> > >
> > > We would like to express our gratitude to the reviewer for raising both the intriguing question and the score.
> > > Consider the example from [19], where the convergence rate is given by $\left(\frac{r_n^{2d}}{n \varepsilon^2}\right)^{\frac{\alpha}{\alpha+d}}$.
> > > When the set $\mathcal{X} = \times_{j=1}^d[a^j,b^j]$ is unknown, it becomes necessary to create a histogram partition over $\times_{j=1}^d[-r_n, r_n]$, introducing an additional factor of $r_n^{2d}$.
> > > However, with publicly available data, we can approximate the range of the $j$-th dimension using $\widehat{a^j} = \min_{i}X_i^j$ and $\widehat{b^j} = \max_{i}X_i^j$.
> > > Subsequently, we can perform min-max scaling on each data point from $\times_{j=1}^d[\widehat{a^j},\widehat{b^j}]$ to map it into $\times_{j=1}^d[0,1]$, and then train an LPDT on $\times_{j=1}^d[0,1]$. Any $x$ that falls outside the range $\times_{j=1}^d[0,1]$ is predicted as $0$.
> > > In the following, we demonstrate that by following this approach, Theorem 3.4 holds with a probability of at least $1-d/n_q^2$ (this is how "With high probability, this is equivalent to..." in the rebuttal can be interpreted).
> > > At present, we assume that $\mathrm{P}_X = \mathrm{Q}_X$, but the analogous conclusion holds as long as their density ratio is bounded.
> > >
> > >
> > > Proof: Let $X$ be scaled to $\breve{X}$.
> > > The excess risk can be decomposed by
> > > $$R_{L, P}\left(f_\pi^{DP}\right)-R_{L, {P}}^*  = \int_{[0,1]^d} L(x,y, f_{\pi}^{DP}(x))dP(x,y) - \int_{[0,1]^d} L(x,y, f*(x))dP(x,y) +  \int_{[0,1]^{dc}} L(x,y, f_{\pi}^{DP}(x))d\mathrm{P}(x,y) - \int_{[0,1]^{dc}} L(x,y, f^{*}(x))dP(x,y) \lesssim  \left(\frac{\log n}{n \varepsilon^2}\right)^{\frac{\alpha}{\alpha+d} \wedge \frac{1}{3}} + M^2 \mathbb{P}(\breve{X}\notin \times_{j=1}^d[0,1]).$$
> > >
> > > Note that
> > > $
> > > \mathbb{P}(\breve{X}\notin \times_{j=1}^d[0,1]) =
> > > \mathbb{P}(X\notin \times_{j=1}^d[\widehat{a^j},\widehat{b^j}]) \leq  \sum_{j=1}^d \mathbb{P}(X^j\notin [\widehat{a^j},\widehat{b^j}]).
> > > $
> > > Denote the CDF of $X^j$ as $F^j$. Since
> > > $
> > > \mathbb{P}\left(\left| F^j(\widehat{a^j}) -F^j(a^j)\right|>\frac{2\log n_q}{n_q}\right) =\mathrm{P}\left(  F^j(\widehat{a^j} ) >\frac{2\log n_q}{n_q}\right)
> > >  = \left(1 - \frac{2\log n_q}{ n_q}\right)^{n_q} = \frac{1}{n_q^2}
> > > $
> > > and similar result holds for $b^j$, we have $\mathbb{P}(X^j\notin [\widehat{a^j},\widehat{b^j}]) \lesssim \frac{\log n_q}{n_q}$
> > > with probability $1- 1/n_q^2$.
> > > Thus, there holds $\mathbb{P}(X\notin \times_{j=1}^d[\widehat{a^j},\widehat{b^j}])) \lesssim \frac{\log n_q}{n_q}$
> > > with probability $1- d/n_q^2$.
> > > Note that if we restrict $n_q \gtrsim (n\varepsilon^2)^{\frac{d \vee 2\alpha }{2 \alpha+2 d}}$ (instead of $n_q \gtrsim n^{\frac{d  }{2 \alpha+2 d}}$ as in the paper, which is a minor change),
> > > there holds $\mathbb{P}(\breve{X}\notin \times_{j=1}^d[0,1]) \lesssim (n\varepsilon^2)^{-\frac{\alpha}{\alpha+d} }\cdot \log n $.
> > > Thus, up to a log factor, Theorem 3.4 holds with probability $1-d/n_q^2$.
> > >
> > > [19] László Györfi and Martin Kroll. On rate optimal private regression under local differential privacy. arXiv preprint arXiv:2206.00114, 2022.

---

> > > > ### Comment · Reviewer_QZtg · 2023-08-16
> > > >
> > > > Thanks for your detailed response. I have one more question. Compared to Theorem 3.4, the failure probability increased from $2/n_q^2$ to $d/n_q^2$, which I believe inherently requires that $n_q>\sqrt{d}$. Would this become a significant drawback?

---

> > > > > ### Author Response · Authors · 2023-08-16
> > > > >
> > > > > Thanks for the question.
> > > > > We believe this is a minor issue for our theoretical results.
> > > > > On one hand, dimension $d$ is generally treated as a constant and considered to be small in the context of nonparametric statistical theory.
> > > > > On the other hand, we can get the same argument with probability $1 - d / n_q^a$ for any $a>0$ by relaxing the upper bound from $\left(n \varepsilon^2\right)^{-\frac{\alpha}{\alpha+d}}\cdot \log n$ to $\frac{a}{2}\left(n \varepsilon^2\right)^{-\frac{\alpha}{\alpha+d}}\cdot \log n$.
> > > > > Thus, with only constant change to the upper bound, we can mitigate the requirement of $n_q$ on $d$ as we like, for instance, to $n_q > d^{1/4}$.

---

> > > > > > ### Comment · Reviewer_QZtg · 2023-08-16
> > > > > >
> > > > > > Thanks! This resolved my question. I will update my score to 6.

---

### Official Review · Reviewer_9jqb · 2023-07-09

**Soundness:** 3 good
**Presentation:** 3 good
**Contribution:** 3 good
**Rating:** 6
**Confidence:** 3

**Summary:**

This paper proposes a locally differentially private (LDP) decision-tree (DT) regressor algorithm that takes advantage of public data for utility improvement.
The proposed algorithm, LPDT, uses Randomized Response (RR) and the Laplace mechanism to protect the tree partition step.
This paper also introduces a new splitting rule named "max-edge partition rule" by using the variance as a reduction criterium.
Theoretical results were presented for convergence rates, training/testing time and space complexity, with important gains in comparison with the state-of-the-art.
Experimental results were provided with synthetic and many real-world datasets to validate the proposed approach, with significative gains over the state-of-the-art.

**Strengths:**

The motivation behind using Locally Differentially Private (LDP) methods for decision trees is well-founded. The idea of using public data is new and has proven to bring advantages in many fields (e.g., vision, languages).
Theoretical and experimental validation were presented showing advancement over the state-of-the-art.

**Weaknesses:**

The experiments only considered medium privacy regimes \epsilon >= 2. The paper would greatly benefit from including high privacy regimes \epsilon <=1 and the utility gain of using public data.

**Questions:**

- Why only presenting results for medium to high privacy regimes?
- In lines 73 & 74 "we show that LPDT performs well even in the presence of significant disparities between public and private data.", it is unclear how "performs well" is concise with the results of Fig. 4. For instance, training only on private date led to MSE=0.8 (Table 3). Using only \alpha=1 as public data, with \epsilon=3 (medium regime), the MSE~3.5 is ~4.5x larger. How worse would that be in high privacy regimes?

**Limitations:**

Although using public data can lead to utility improvements, the authors did not discuss the limitation of having such data in real-world applications. Indeed, following my last question on "data heterogeneity", the learning process could be really damaged if public and private data have different distributions. I recommend the authors to further discuss this paper's limitations.

---

> ### Author Rebuttal · Authors · 2023-08-10
>
> Q1: The paper would greatly benefit from including high privacy regimes $\epsilon$ <=1 and the utility gain of using public data.
>
> A1: Please see R1 of the global response for small $\varepsilon$ and Experiment 4 for the utility gain.
>
> Q2: In lines 73 \& 74 ... How worse would that be in high privacy regimes?
>
> A2:
> We apologize for the unclearness of our statement.
> The conclusion aims to state that the overhead brought by the distribution shift is minor compared to the overhead brought by privacy.
> For instance, using $\varepsilon = 4$ and $\delta = 0$ results an MSE ~2.3, which increased 1.5 compared to  $\varepsilon = \infty$ and $\delta = 0$.
> Yet, using $\varepsilon = 3$ and $\delta = 1$ leads to an MSE ~2.9, which is ~0.6 higher.
> The results of the high privacy regime are in Experiment 2 in the global response.
>
>
>
>
>
> Q3: Although using public data can lead to utility improvements, the authors did not discuss the limitation of having such data in real-world applications. Indeed, following my last question on "data heterogeneity", the learning process could be really damaged if public and private data have different distributions.
>
>
>
> A3:
> Please see R2 in the global response for a discussion of public data.
> We will add the discussion of the limitations.

---

> > ### Comment · Reviewer_9jqb · 2023-08-16
> > **Thank you for detailed rebuttal**
> >
> > Dear authors, thank you for your detailed response to all reviewers' comments. I have updated my score to 6.

---

### Author Rebuttal · Authors · 2023-08-10

We express our gratitude to the reviewers for their insightful questions and valuable comments. Reviewers are encouraged to refer to their individual rebuttals, where we direct them to the relevant sections in the text provided below.

R1: We did not incorporate the results of $\varepsilon < 1$ due to the following two principal reasons:

1. Challenges in High Privacy Regime: The LDP nonparametric regression problem becomes particularly challenging in a high privacy regime. Consequently, model performances tend to resemble random noise, making them difficult to distinguish. When $\varepsilon < 1$, model performance degrades significantly, as shown in [4]. Additionally, nonparametric regression presents greater complexity compared to more commonly encountered LDP problems such as heavy hitters. Nonparametric regression involves estimating both the joint density and the marginal density, each utilizing a privacy budget of $\varepsilon / 2$.

2. The mild privacy budgets $\varepsilon > 1$ meet practical needs. For instance, Apple conduct most tasks under $\varepsilon \geq 2$ [8]. Also, for LDP nonparametric regression, prior empirical studies have exclusively focused on scenarios where $\varepsilon > 1$ [5].

We do have the experimental results for the high privacy regime in R3.

R2: We summarize how the estimator benefits from public data, both theoretically and empirically.

1. Theoretically, the advantages of LPDT lie in two aspects: i) LPDT allows the marginal density to be arbitrarily low with no truncation procedure. ii) LPDT eliminates the range parameter that appeared in the previous work [1,7]. This is not emphasized in the paper. The range parameter appears due to the unknown support $\mathcal{X}$. But with public data, we can min-max scale $X$ to $[0,1]^d$. With high probability, this is equivalent to working under known $\mathcal{X} = [0,1]^d$.  From a theoretical perspective, LPDT benefits from the public data through marginal distribution only.

2. Empirically, the labels of public data can boost performance by creating more informative partitions. When the regression function of $\mathrm{P}$ and $\mathrm{Q}$ are identical, LPDT creates similar partitions as in the non-private case. When the regression function of $\mathrm{P}$ and $\mathrm{Q}$ have similar trends but are distinct (e.g. when $Y$ are monotonic with respect to each $X$ in both $\mathrm{P}$ and $\mathrm{Q}$), the partition created on public data can be informative for private estimation. However, employing public data in the estimation process will result in asymptotically biased estimation. In cases where the regression function of $\mathrm{P}$ and $\mathrm{Q}$ are completely different, using public data to create partitions is non-informative.

3. Whether public data is advantageous or not can often be intuitively determined based on the data source. For example, a website might consider the data of users who willingly share their information as public data, as the behaviors of users who choose to share information or not are similar. Quantitatively determining whether a dataset can be utilized as public data can be of independent interest, as demonstrated by [6].

R3: The additional experiment results with tables and plots in the PDF.

1. Experiment 1: We replicate the same experiment outlined in Section 4.2 with two privacy parameters, namely $\varepsilon = 0.5$ and $\varepsilon = 1$. The results are shown in Table 1. Under high privacy constraints, both LPDT-M and LPDT-V consistently outperform their competitors. However, it's noteworthy that the performance of all models is worse than the optimal lower bound (decision tree) for $\varepsilon = 0.5$. For some results, the deviation from the optimal lower bound increases even by magnitudes.

2. Experiment 2: We replicate the same experiment conducted on taxi data with $\varepsilon = 0.5, 1, 2$. The results are in Table 2.
 As anticipated, the performance of LPDT is notably compromised for small $\varepsilon$ values. Specifically, when $\epsilon < 3$, the performance of LPDT is inferior to training a decision tree directly on public data. However, it's important to note that LPDT still demonstrates superior performance compared to other LDP methods.

3. Experiment 3: We replicate the privacy utility trade-off experiment on the synthetic dataset with extended privacy regime $\varepsilon = 0.5, 1, 2$. The results are in Figure 1(a). In the high privacy regime, LPDT, PHIST, and APHIST exhibit comparable performance.

4. Experiment 4: We conduct experiments to show the utility gain brought by public data. We take red wine data as an example. With $n_q = 100$ public data, we run LPDT (with $D^{\mathrm{P}}$) and PHIST (without $D^{\mathrm{P}}$). For $\delta \in \{0.2, 0.4, 0.6, 0.8,1\}$, we train each model on $\delta \cdot n$ samples with 20 repetations. The result is in Figure 1(b). PHIST with ~1,100 samples achieves the same MSE as LPDT with ~660 samples. The result shows that, with a small amount of public data, LPDT achieves the same performance with much fewer samples.

[1] Thomas B Berrett et all. Strongly universally consistent nonparametric regression and classification with privatised data. 2021.

[2] T Tony Cai et al. Transfer learning for nonparametric regression: Non-asymptotic minimax analysis and adaptive procedure. 2022.

[3] T Tony Cai et al. Transfer learning for nonparametric classification: Minimax rate and adaptive classifier. 2021.

[4] John C Duchi et al. Minimax optimal procedures for locally private estimation. 2018.

[5] Farhad Farokhi. Deconvoluting kernel density estimation and regression for locally differentially private data. 2020.

[6] Xin Gu et al. Choosing public datasets for private machine learning via gradient subspace distance. 2023.

[7] László Györfi et al. On rate optimal private regression under local differential privacy. 2022.

[8] Apple Inc. Differential privacy technical overview. Technical Report Apple Inc., 2017.

---

### Decision · Program_Chairs · 2023-09-21

**Decision:**

Accept (poster)

**Comment:**

Reviewers remained mum during the discussion, so I took a look at the reviews and responses myself. It seems like most of the weaknesses identified by reviewers were comparatively minor, and the authors did more experiments for some of the gaps identified by the reviewers. These additional results and experiments should appear in the final paper. Overall, since all reviewers generally had a positive sentiment, this paper is appropriate for publication at NeurIPS.